# Co-agonists differentially tune GluN2B-NMDA receptor trafficking at hippocampal synapses

Joana S Ferreira[1,2†], Thomas Papouin[2,3†‡], Laurent Ladépêche[1,2], Andrea Yao[4], Valentin C Langlais[2,3], Delphine Bouchet[1,2], Jérôme Dulong[2,3], Jean-Pierre Mothet[5], Silvia Sacchi[6,7], Loredano Pollegioni[6,7], Pierre Paoletti[4], Stéphane Henri Richard Oliet[2,3], Laurent Groc[1*]

[1]Interdisciplinary Institute for NeuroSciences, CNRS UMR 5297, Bordeaux, France; [2]Université de Bordeaux, Bordeaux, France; [3]NSERM U862, Neurocentre Magendie, Bordeaux, France; [4]Institut de Biologie de l'ENS (IBENS), CNRS UMR 8197, INSERM U1024, Paris, France; [5]Université Aix-Marseille, CNRS CRN2M UMR 7286, Marseille, France; [6]Dipartimento di Biotecnologie e Scienze della Vita, Università degli Studi dell'Insubria, Varese, Italy; [7]The Protein Factory, Centro Interuniversitario di Biotecnologie Proteiche, Politecnico di Milano, Università degli Studi dell'Insubria, Varese, Italy

*For correspondence: laurent. groc@u-bordeaux.fr

†These authors contributed equally to this work

Present address: ‡Neuroscience department, Tufts University School of Medicine, Boston, United States

**Abstract** The subunit composition of synaptic NMDA receptors (NMDAR), such as the relative content of GluN2A- and GluN2B-containing receptors, greatly influences the glutamate synaptic transmission. Receptor co-agonists, glycine and D-serine, have intriguingly emerged as potential regulators of the receptor trafficking in addition to their requirement for its activation. Using a combination of single-molecule imaging, biochemistry and electrophysiology, we show that glycine and D-serine relative availability at rat hippocampal glutamatergic synapses regulate the trafficking and synaptic content of NMDAR subtypes. Acute manipulations of co-agonist levels, both ex vivo and in vitro, unveil that D-serine alter the membrane dynamics and content of GluN2B-NMDAR, but not GluN2A-NMDAR, at synapses through a process requiring PDZ binding scaffold partners. In addition, using FRET-based FLIM approach, we demonstrate that D-serine rapidly induces a conformational change of the GluN1 subunit intracellular C-terminus domain. Together our data fuels the view that the extracellular microenvironment regulates synaptic NMDAR signaling.

## Introduction

Glutamatergic synapses mediate most excitatory neurotransmission in the brain, predominantly through the activation of ionotropic glutamate receptors, such as the NMDA receptor (NMDAR). In addition to glutamate, the NMDAR activation requires the binding of a co-agonist. It has been established that endogenous glycine and D-serine can both play thid ms role (*Oliet and Mothet, 2009*) even though D-serine was shown to be the endogenous co-agonist at glutamatergic synapses in the adult forebrain (*Papouin et al., 2012*). Most NMDAR are heterotetramers comprising various combinations of GluN1 (binding the co-agonist) and GluN2A-D (binding glutamate) subunits, which confer specific biophysical, pharmacological, and signaling properties to the receptor (*Paoletti et al., 2013*). In particular, a variety of different properties and functions have been attributed to GluN2A-containing and GluN2B-containing NMDAR subtypes (thereafter referred to as GluN2A- and GluN2B-NMDAR), such as their alleged roles in plasticity (*Paoletti et al., 2013*). During

development, the NMDAR composition is tightly regulated: GluN2B subunits are highly expressed during early development and reach a peak around the second postnatal week, whereas GluN2A subunit levels increase only after birth, exceeding GluN2B subunits by adulthood (*Monyer et al., 1994*; *Barth and Malenka, 2001*). The progressive enrichment of GluN2A-NMDAR at developing synapses is essential for synaptic maturation, neuronal network and cortical map establishment (*Yashiro and Philpot, 2008*).

It recently emerged that synaptic NMDAR are mostly gated by D-serine, whereas extrasynaptic receptors are gated by glycine (*Papouin et al., 2012*; *Sullivan and Miller, 2012*). This co-agonist segregation is accompanied by a similar compartmentalization of NMDAR subtypes wherein synaptic NMDAR were mostly GluN2A-NMDAR, whereas GluN2B-NMDAR were found at extrasynaptic locations. This raised the question of whether the distribution of NMDAR subtypes on CA1 neurons is responsible for the spatial segregation of the co-agonist or, on the contrary, whether the compartmentalization of glycine and D-serine dictates the location of NMDAR subtypes on hippocampal neurons. Although there is little evidence that NMDAR subtypes have a preferred co-agonist (*Priestley et al., 1995*; *Wafford et al., 1995*; *Chen et al., 2008*), it appears that glycine and D-serine differentially regulate the surface dynamics of NMDAR on hippocampal neurons (*Burnet et al., 2011*; *Papouin et al., 2012*). In addition, both glycine and D-serine alter NMDAR internalization in a clathrin-dependent mechanism (*Nong et al., 2003*). Here, we thus investigated whether the two NMDAR co-agonists, glycine and D-serine, modulate the GluN2A/B subunit composition at hippocampal synapses, using a unique combination of single-molecule imaging, immunocytochemistry, biochemistry and electrophysiology in hippocampal networks. We found that the relative abundance of glycine and D-serine dictates GluN2B-NMDAR content in synapses. Since NMDAR surface dynamics modulate NMDAR-dependent synaptic signaling and plasticity (*Groc et al., 2006*; *Dupuis et al., 2014*), it thus appears that NMDAR co-agonists act as major regulators of the NMDAR function through subtype-specific alterations of receptor trafficking.

## Results

### GluN2B-NMDAR surface dynamics and distribution are differentially altered by D-serine and glycine

Using electrophysiological and single-molecule imaging approaches, it emerged that the NMDAR synaptic content relies on both receptor cycling with intracellular pool and fast membrane lateral diffusion (*Bard and Groc, 2011*). To explore the impact of D-serine or glycine on endogenous NMDAR trafficking, we performed single nanoparticle (Quantum Dot, QD) tracking experiments in cultured hippocampal networks (*Figure 1A*). These were performed in 10–18 days in vitro (div)-old neurons, a time at which glutamatergic synapses express both GluN2A- and GluN2B-NMDAR, have spine-like morphology, and naturally contain both glycine and D-serine (*Groc et al., 2006*; *Bard et al., 2010*; *Rosenberg et al., 2013*). To assess the effect of glycine and D-serine on receptor surface dynamics, we used exogenous application of each co-agonist or specific enzymatic scavengers (*Pollegioni et al., 1992*; *Job et al., 2002*; *Papouin et al., 2012*) that selectively degrade endogenous extracellular D-serine (D-amino acid oxidase, *Rg*DAAO) or glycine (glycine oxidase, *Bs*GO). The control condition refers to the situation before stimuli. We found that the instantaneous diffusion coefficient of GluN2A-NMDAR was decreased by 14% in presence of glycine, whereas D-serine had no effect (glycine: 86 ± 1% of control, $n$ = 8115 trajectories, p<0.0001; *Figure 1b–c*; D-serine: 100 ± 2% of control, $n$ = 5555 trajectories, p>0.05; *Figure 1B–C*). In order to examine the diffusion pattern of GluN2A-NMDAR (e.g. confined or freely diffusive), we plotted the mean square displacement (MSD) versus time lag, which represents the average area explored by the receptor over time, upon exogenous application of co-agonists. We found that the MSD curves of GluN2A-NMDAR in the presence of glycine or D-serine were undistinguishable (glycine: $MSD^{0.5-0.75s}$ = 0.2135 ± 0.011 $\mu m^2$; D-serine: $MSD^{0.5-0.75s}$ = 0.246 ± 0.012 $\mu m^2$; $n$ = 250–252 selected trajectories for each group, p=0.1087; *Figure 1D*). In contrast, GluN2B-NMDAR surface diffusion was markedly reduced in presence of exogenous D-serine, whereas glycine incubation was without significant effect (D-serine: 57 ± 11% of control, $n$ = 836 trajectories, p<0.001; *Figure 1E–G*; glycine: 71.5 ± 11% of control, $n$ = 318 trajectories, p>0.05; *Figure 1E–F*). In addition, the diffusion pattern of GluN2B-NMDAR became markedly confined in the presence of D-serine but not glycine (D-serine: $MSD^{0.5-}$

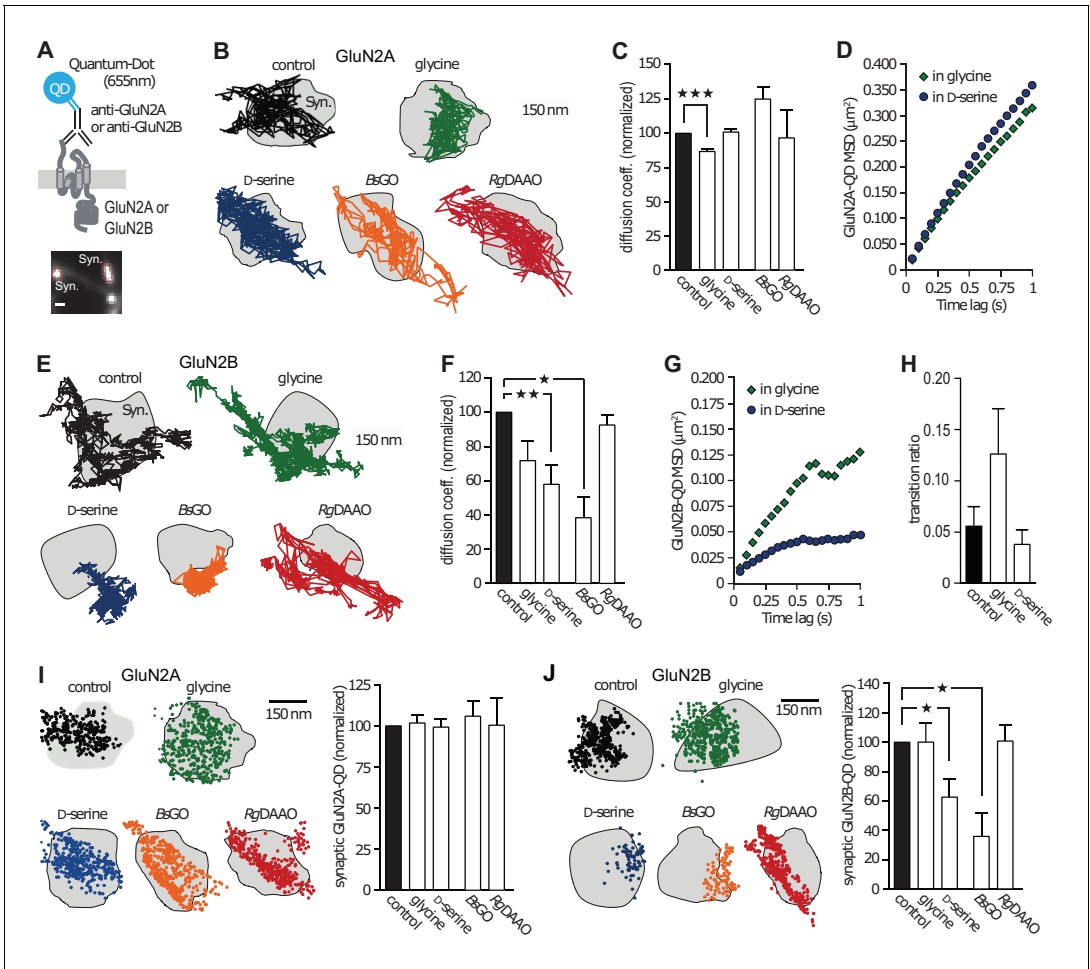

**Figure 1.** GluN2B-NMDAR synaptic dynamics is specifically altered by D-serine. (**A**) Diagram representing the Quantum Dot (QD) coupled to the specific antibody used to track single surface GluN2-NMDAR (top). Example of a synapse area (Syn) identified after incubation with MitoTracker (bottom). Scale: 1 µm. (**B**) Examples of the surface trajectories of single QD-coupled GluN2A-NMDAR in the synaptic area. Scale: 150 nm. (**C**) Mean diffusion coefficient of synaptic GluN2A-QD normalized to control condition (before co-agonist or enzyme application). Glycine: $n = 8,115$, D-serine: $n = 5,555$, $Bs$GO: $n = 308$, $Rg$DAAO: $n = 121$ trajectories; $p<0.0001$ Kruskal-Wallis test. (**D**) Mean Square Displacement (MSD) of surface GluN2A trajectories measured either on the presence of glycine or D-serine. (**E**) Examples of the surface trajectories of single QD-coupled GluN2B-NMDAR as in (**B**). (**F**) Mean diffusion coefficient of synaptic GluN2B-QD normalized to control condition. Glycine: $n = 318$, D-serine: $n = 836$, $Bs$GO: $n = 73$, $Rg$DAAO: $n = 605$ trajectories; $p<0.0001$ Kruskal-Wallis test. (**G**) MSD of surface GluN2B trajectories as in (**D**). (**H**) Transition ratio (i.e. entries or exits from synaptic areas/total number of trajectories per cellular field) of GluN2B-QD in the presence of either glycine or D-serine. Control: $n = 7$, glycine: $n = 4$, D-serine: $n = 7$ neuronal fields; $p=0.1187$ Kruskal-Wallis test. (**I–J**) Synaptic fraction of QD-detected GluN2A-NMDAR (**I**) and GluN2B-NMDAR (**J**) in the synaptic area, normalized to the respective controls. GluN2A: $n = 21$ glycine, $n = 21$ D-serine, $n = 16$ BsGO, $n = 22$ $Rg$DAAO dendritic fields; $p=0.9974$ Kruskal-Wallis test. GluN2B: $n = 8$ glycine, $n = 14$ D-serine, $n = 37$ BsGO, $n = 15$ $Rg$DAAO dendritic fields; $p<0.0001$ Kruskal-Wallis test. Data are represented as mean ± s.e.m.; ***$p<0.0001$, **$p<0.001$, *$p<0.05$, Dunn's Multiple Comparison Test.

$0.75s = 0.042 \pm 0.0005$ µm$^2$; glycine: MSD$^{0.5-0.75s} = 0.107 \pm 0.003$ µm$^2$; $n = 252$ selected trajectories for each group; $p=0.0022$; *Figure 1G*). These changes were not accompanied by alteration of the exchange rate between synaptic and extrasynaptic compartments (control: $0.056 \pm 0.019\%$, $n = 7$ neuronal fields, glycine: $0.126 \pm 0.044\%$, $n = 4$ neuronal fields, D-serine: $0.038 \pm 0.015\%$, $n = 7$ neuronal fields, $p=0.1187$; *Figure 1H*).

To investigate whether the endogenous levels of co-agonists also shape the basal trafficking of NMDAR, we used enzymatic scavengers that specifically degrade D-serine ($Rg$DAAO) or glycine ($Bs$GO). We found that neither $Bs$GO nor $Rg$DAAO impacted surface dynamics of GluN2A-NMDAR ($Bs$GO: $124 \pm 8.5\%$ of control, $n = 308$ trajectories; $Rg$DAAO: $96 \pm 20\%$ of control, $n = 121$ trajectories, $p>0.05$; *Figure 1C*). In contrast, reducing endogenous glycine levels with $Bs$GO decreased

GluN2B-NMDAR diffusion coefficient (*Bs*GO: 37.5 ± 12.5% of control, *n* = 73 trajectories, p<0.05; *Figure 1F*). The degradation of endogenous D-serine with *Rg*DAAO, on the other hand, had no effect on GluN2B-NMDAR diffusion coefficient, much like glycine application (*Rg*DAAO: 92 ± 5.5% of control, *n* = 605 trajectories, p>0.05; *Figure 1F*). Together, these results indicate that under conditions in which D-serine prevails, the diffusion of GluN2B-NMDAR is slowed down and the receptors exhibit a more confined behavior, an effect possibly due to altered interactions with synaptic partners (*Groc et al., 2006*; *Bard et al., 2010*). Consistently, the relative content of synaptic GluN2B-QD, estimated as the synaptic fraction of single GluN2B-QD, was strongly reduced by D-serine incubation and degradation of glycine (D-serine: 62.5 ± 12% of control, *n* = 14 dendritic fields, p<0.05; *Bs*GO: 36 ± 15% of control, *n* = 37 dendritic fields, p<0.05; *Figure 1J*). This effect was specific to GluN2B-QD as the density of GluN2A-QD remained unaffected (n = 12–21 dendritic fields, p>0.05; *Figure 1I*).

## Synaptic content of GluN2B-NMDAR is reduced by exogenous application of D-serine

The reduced number of GluN2B-QDs at synaptic areas suggests that the D-serine-induced decrease in GluN2B-NMDAR trafficking lead to a concomitant decrease of the receptor synaptic content. To test this, we assessed GluN2-NMDAR synaptic content in cultured hippocampal neurons using live immunocytochemistry with specific antibodies against the extracellular N-terminal of endogenous GluN2A or GluN2B subunit. The synaptic compartment was identified by the presence of the post-synaptic protein Homer-1c and the synaptic localization of GluN2A- and GluN2B-NMDAR was determined by the extent of co-localization of their surface expression with this marker. Incubation with glycine or D-serine (45 min, 30 µM) did not alter the synaptic content of GluN2A-NMDAR (113 ± 12% and 100 ± 8% of control for glycine and D-serine, respectively, *n* = 39–41 cells, p>0.05; *Figure 2A,B*) and the total (extra- and synaptic) surface amount of clusters (110 ± 9% and 103 ± 7% of control for glycine and D-serine, respectively, p>0.05; *Figure 2A,B*). Using the same anti-GluN2A subunit antibody under permeabilized condition, to stain both intracellular and surface GluN2A-NMDAR, revealed no change in the total content (glycine: 97 ± 5%, D-serine 105 ± 5% compared to control, respectively, p>0.05, *Figure 2C*). In contrast, D-serine reduced the surface synaptic staining of GluN2B-NMDAR (68 ± 7% of control, *n* = 39–40 cells, p<0.05; *Figure 2D,E*), whereas glycine was without significant effect (79 ± 7% of control, *n* = 40 cells, p>0.05; *Figure 2E*). This was specific to synaptic clusters since the surface staining (extra- and synaptic) of GluN2B-NMDAR was unaffected in all conditions (glycine: 91.5 ± 7%, D-serine: 82 ± 7%, compared to control, *Figure 2E*). The total content of GluN2B-NMDAR, evaluated by the labeling of GluN2B subunits after permeabilization, was not altered by glycine or D-serine incubation (glycine: 103 ± 9%, D-serine: 112 ± 8% compared to control, *Figure 2F*). Furthermore, the total content of GluN1 subunit (staining in permeabilized condition) was unaltered by glycine or D-serine incubation (glycine: total 102 ± 8%, synaptic 101 ± 8%; D-serine: total 94 ± 5%, synaptic 88 ± 5% compared to control, *Figure 2—figure supplement 1*). These results are thus consistent with the single GluN2B-NMDAR-QD distributions described above, supporting the claim that GluN2B-NMDAR surface diffusion and synaptic content are altered by D-serine. To confirm these findings in a more intact preparation, synaptosomes were isolated from hippocampi of young adult rats (P30) and incubated 1 hr with either co-agonist (30 µM, *Figure 2G*). Then, GluN1, GluN2A and GluN2B subunit contents were quantified by Western blotting. Consistently, the GluN2B, but not GluN2A, subunit level was significantly decreased after D-serine incubation (glycine/D-serine ratio change relative to control, GluN1 1 ± 10%, GluN2A 10 ± 8%, GluN2B 28 ± 5%, *n* = 5 independent experiments, *Figure 2H–I*). Also, the total amount of GluN2B subunit was not altered by D-serine incubation (control: 1.0 ± 0.11, *n* = 5 independent experiments; D-Serine: 0.95 ± 0.16, *n* = 5; glycine: 0.71 ± 0.14, *n* = 5; p>0.05, ANOVA one followed by Newman-Keuls multiple comparisons test), supporting a redistribution of GluN2B-NMDAR. Finally, the synaptic content of the obligatory GluN1 subunit was unaffected by either treatment, suggesting a stable amount of NMDAR with thus possible redistributions of other NMDAR subunits (e.g. GluN3A/B).

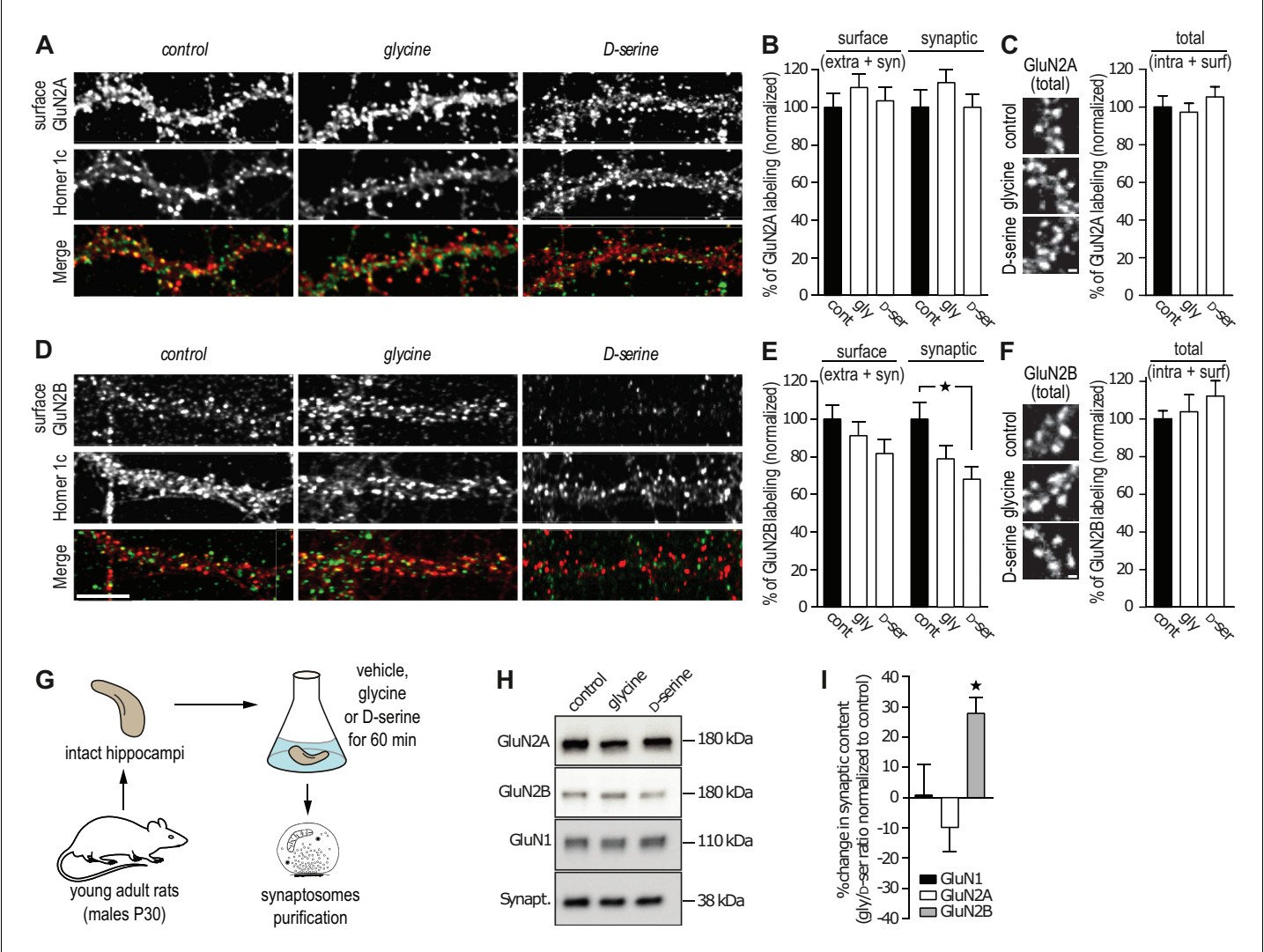

**Figure 2.** GluN2B synaptic content is decreased after D-serine incubation. (A) Surface labeling of GluN2A in hippocampal neurons 18–19 div in culture stimulated with glycine or D-serine. Homer 1c staining was used as the synaptic maker. Scale: 10 μm. (B) Surface (extra- and synaptic) and synaptic GluN2A clusters overlapping Homer 1c positive clusters, normalized to the respective control conditions. Control: n = 41, glycine: n = 39, D-serine: n = 40 cells; surface GluN2A p=0.8935, synaptic GluN2A p=0.8830, Kruskal-Wallis test. (C) Total (intracellular and surface) GluN2A clusters from hippocampal neurons 17 div, labeled after permeabilization, stimulated with glycine or D-serine and normalized to the control. Control: n = 30, glycine n = 30, D-serine n = 30 cells; p=0.6884, Kruskal-Wallis test. (D) Surface labeling of GluN2B in hippocampal neurons 18–19 div in culture stimulated with glycine or D-serine. Homer 1c staining was used as the synaptic maker. Scale: 10 μm. (E) Surface (extra- and synaptic) and synaptic GluN2B clusters overlapping Homer 1c positive clusters, normalized to the respective control conditions. Control: n = 41, glycine: n = 39, D-serine: n = 40 cells; surface GluN2B p=0.1383, synaptic GluN2B p=0.0247, Kruskal-Wallis test followed by Dunn's Multiple Comparison Test, *p<0.05. (F) Total (intracellular and surface) GluN2B clusters from hippocampal neurons 17 div, labeled after permeabilization, stimulated with glycine or D-serine and normalized to the control. Control: n = 30, glycine n = 30, D-serine n = 30 cells; p=0.5478, Kruskal-Wallis test. (G) Synaptosomes were purified by subcellular fractionation from rats P30 hippocampi incubated for 60 min in aCSF containing either glycine or D-serine. (H) 1 μg of protein was probed against GluN2B, GluN2A and GluN1. Synaptophysin was used as a loading control. (I) Synaptic fraction of NMDAR subunits levels, calculated as the variation between the protein expression levels in glycine and in D-serine conditions, normalized to the non-treated condition, control. n = 5, GluN1 levels p=0.4409, GluN2A levels p=0.1062, GluN2B levels p=0.0207, Repeated measures ANOVA followed by Bonferroni's multiple comparison test, *p<0.05. Data are represented as mean ± s.e.m.

The following figure supplement is available for figure 2:

**Figure supplement 1.** GluN1 content is not altered by the co-agonists application.

# Co-agonist-dependent regulation of GluN2B-NMDAR trafficking depends on interaction between NMDAR and PDZ scaffolds

Scaffolding proteins within the postsynaptic density are major regulators of GluN2-NMDAR trafficking and localization (*Bard and Groc, 2011*). The C-terminus of GluN2 subunits is a well-known hub for interactions with several intracellular partners, especially PDZ scaffold proteins such as PSD-95 (*Elias and Nicoll, 2007*). Since our results point to a robust alteration of GluN2B-NMDAR synaptic trafficking and content in presence of D-serine, we hypothesized that this effect depend on an interaction between intracellular scaffold proteins and the C-terminus of the GluN2B subunit. To test this possibility, we prevented such interaction using a competing peptide and reasoned that this should prevent the effect of D-serine on GluN2B-NMDAR trafficking. We used a TAT-GluN2B$_{15}$ peptide mimicking the last 15 amino acids of the GluN2B subunit, to efficiently disrupt the interaction between GluN2B-NMDAR and PDZ proteins, as previously demonstrated (*Bard et al., 2010*), and a scramble peptide as control (TAT-NS, 20 µM). In presence of TAT-NS (15 min pre-incubation), we observed the expected decrease in GluN2B-NMDAR diffusion elicited by D-serine application (90.5 ± 6.3% of control, *n* = 229 trajectories, p<0.05; *Figure 3A–B*). This was accompanied by an increased confinement behavior of GluN2B-NMDAR (*Figure 3C*, TAT-NS: MSD$^{0.35\text{-}0.55s}$ = 0.1929 ± 0.005 µm$^2$, TAT-NS + D-serine: MSD$^{0.35\text{-}0.55s}$ = 0.1625 ± 0.002 µm$^2$, p=0.0079). Remarkably, both these effects were prevented in the presence of the TAT-GluN2B$_{15}$ peptide: D-serine no longer altered the diffusion coefficient (101.1 ± 4.03% of control TAT-GluN2B, *n* = 713 trajectories, p>0.05; *Figure 3A–B*), or changed the confinement behavior of GluN2B-NMDAR (*Figure 3C*, TAT-GluN2B: MSD$^{0.35\text{-}0.55s}$ = 0.1802 ± 0.003 µm$^2$, TAT-GluN2B + D-serine: MSD$^{0.35\text{-}0.55s}$ = 0.1740 ± 0.004 µm$^2$, p=0.4206). As previously observed, D-serine decrease the synaptic content of GluN2B-NMDAR (co-localized with Homer-1c) in neurons exposed 45 min to the TAT-NS peptide (85.3 ± 2.5% of control TAT-NS, *n* = 38 neuronal fields; *Figure 3D–E*); however, this effect was prevented by TAT-GluN2B$_{15}$ peptide incubation (94.7 ± 2.6% of control TAT-GluN2B, *n* = 70 neuronal fields; *Figure 3D–E*). Since the TAT peptide itself appears to attenuate the D-serine effect (from 43% decrease, *Figure 1F*, to a 10% decrease of GluN2B-NMDAR diffusion, *Figure 2B* and 32% decreased synaptic content, *Figure 2E*, to a 15% decrease, *Figure 3E*), we evaluated the effect of a lower concentration of peptides (2 µM). First, we confirmed that this concentration was sufficient to decrease the binding between GluN2B-NMDAR and PSD-95, one of the scaffold PDZ-containing proteins within glutamate synapses. Following a 45 min incubation with TAT-NS or TAT-GluN2B$_{15}$ (2 µM), the number of GluN2B surface clusters that colocalize with PSD-95 clusters was reduced by the TAT-GluN2B$_{15}$ peptide (TAT-NS incubation, 7.5 ± 0.5 clusters per 10 µm, n = 30 cells, three independent experiments; TAT-GluN2B$_{15}$: 5.8 ± 0.3 GluN2B clusters per 10 µm of dendrite, *n* = 30 cells; p=0.0091, t-test). Then, we investigated the effect of D-serine in presence of this low concentration of peptides. We report a similar extent of GluN2B-NMDAR decrease after D-serine incubation when compared to the higher TAT concentration (TAT-NS 2 µM + D-serine: 90 ± 4% compared to TAT-NS without D-serine, n = 10 cells per condition), suggesting that TAT peptide, even at low concentration, attenuate the D-serine effects on NMDAR trafficking. Together, these data indicate that D-serine binding regulates GluN2B-NMDAR surface trafficking through a C-terminus-mediated process.

This result suggests that the co-agonist availability can modulate the intracellular interaction between the GluN2B subunit and major scaffold partners (e.g. PSD-95 and SAP-102). To identify the main intracellular partner involved in this interaction, GluN2B- and GluN2A-NMDAR complexes were immunoprecipitated from synaptosomes using specific antibodies (*Figure 3F–G*). A subsequent Western-blot quantification analysis revealed that the presence of D-serine, but not glycine, decreased the amount of PSD-95 co-immunoprecipitated with GluN2B-NMDAR (glycine: 100 ± 4.9%, *n* = 10; D-serine: 81.6 ± 2.7% *n* = 8, p<0.05; *Figure 3F*). To note, there was no significant difference between control (no agonist) and glycine condition (control: 2.2 ± 0.4, *n* = 4 independent experiments; glycine: 1.9 ± 0.2, *n* = 4; p>0.05, *t*-test). The effect of D-serine was specific to PSD-95 since its application did not alter the levels of SAP-102, another MAGUK family protein, co-immunoprecipitated with GluN2B-NMDAR (glycine: 100 ± 9.3%; D-serine: 95.5 ± 6.4% *n* = 10, p>0.05; *Figure 3F*), suggesting that co-agonists do not weaken interactions with all members of the PDZ scaffold protein families. There was no effect of glycine or D-serine on the interaction between PSD-95 and GluN2A-NMDAR (glycine: 100 ± 9.7%; D-serine: 98.1 ± 10.3% *n* = 8, p>0.05, *Figure 3G*).

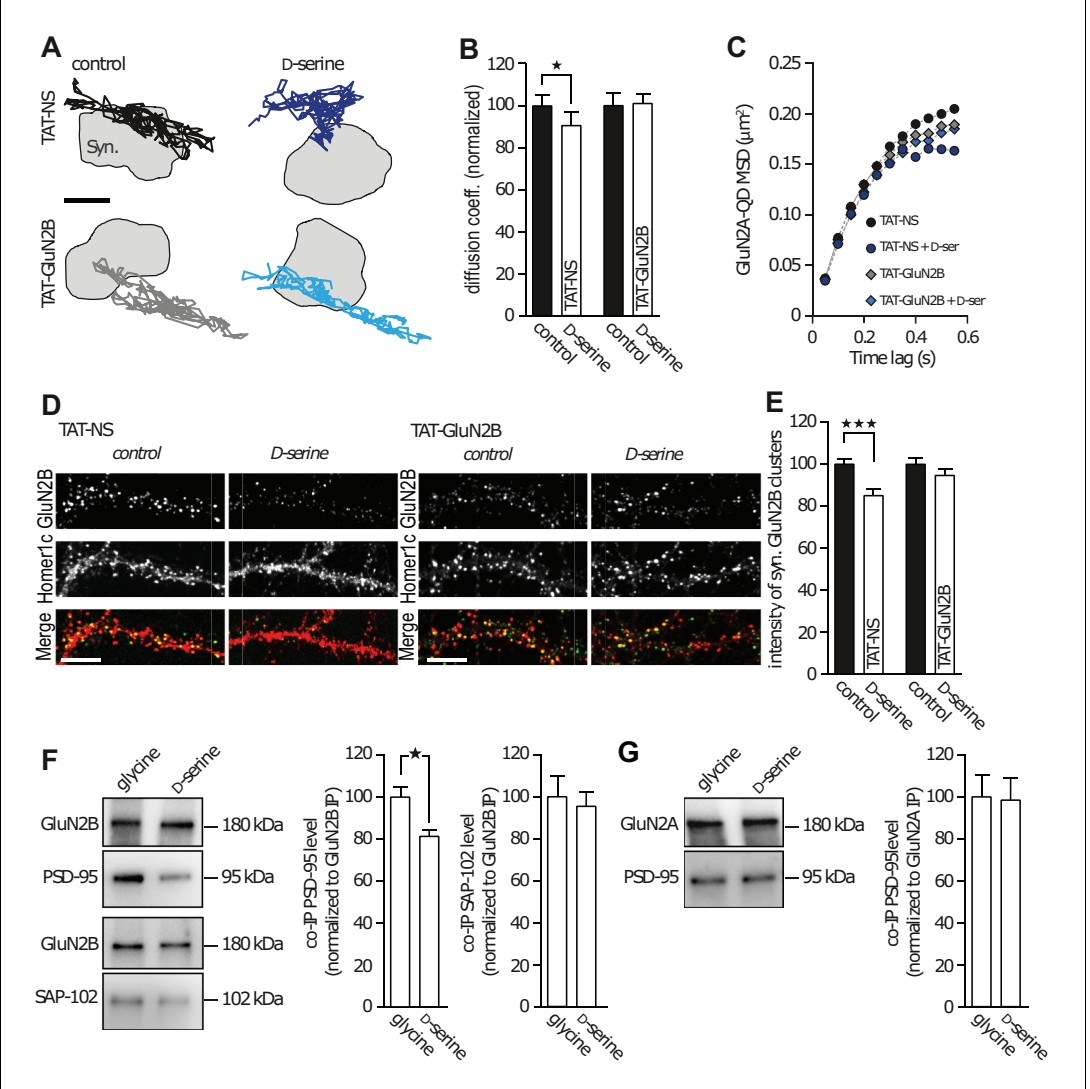

**Figure 3.** GluN2B subunit PDZ-binding domain mediates subunit co-agonist action. (**A**) Examples of the surface trajectories of single QD-coupled GluN2B-NMDAR in the synaptic area. Pre-incubated with scramble peptide (TAT-NS, top) or GluN2B PDZ-proteins disrupting binding (TAT-GluN2B, bottom), before (control, left) or after D-serine application (right). Scale: 150 nm. (**B**) Mean diffusion coefficient of synaptic GluN2B-QD normalized to respective control condition (before D-serine application). TAT-NS control: n = 543, TAT-NS + D-serine: n = 299, TAT-GluN2B control: n = 413, TAT-GluN2B + D-serine: n = 713 trajectories; TAT-NS p=0.0398, TAT-GluN2B p=0.9292, Mann-Whitney test. (**C**) MSD of surface GluN2B trajectories in the presence of either TAT-NS or TAT-GluN2B, with or without D-serine. (**D**) Surface labelling of anti-GluN2B in hippocampal neurons 16–17 div in culture after incubation with either TAT-NS (top) or TAT-GluN2B (bottom), with (right) or without D-serine (left). Homer 1c staining was used as the synaptic maker. Scale: 10 μm. (**E**) Mean intensity of synaptic surface GluN2B clusters (juxtapose to Homer1c clusters) normalized to the respective control. TAT-NS control: n = 41, TAT-NS + D-serine: n = 38, TAT-GluN2B control: n = 61, TAT-GluN2B + D-serine: n = 70 cells; TAT-NS p<0.0001, TAT-GluN2B p=0.3090, Mann-Whitney test. (**F–G**) GluN2B immunoprecipitates (GluN2B IP, (**F**) or GluN2A (GluN2A IP, (**G**) were probed against PSD-95 (top) or SAP-102 (bottom). PSD-95 and SAP-102 co-immunoprecipitation levels (co-IP) normalized to GluN2B (**F**) or GluN2A (**G**) levels after glycine or D-serine incubation. Glycine: n = 10, D-serine: n = 8; PSD-95 co-IP GluN2B p=0.0148; glycine and D-serine: n = 10; SAP-102 co-IP GluN2B p=0.9118; glycine and D-serine: n = 8; PSD-95 co-IP GluN2A p=0.8785, Mann-Whitney test. Data are represented as mean ± s.e.m.; *p<0.05, ***p<0.0001.

These data indicate that D-serine binding specifically reduces the trafficking and synaptic content of GluN2B-NMDAR through altered binding to PDZ containing proteins such as PSD95.

## D-serine alters the conformation of NMDAR C-terminus

It has recently been shown that the agonist binding to the NMDAR produces conformational changes of the C-terminus (*Dore et al., 2015*) and that these changes modulate the binding to several intracellular partners (*Doré et al., 2014*; *Aow et al., 2015*), including PDZ-containing proteins like PSD-95 (*Doré et al., 2014*). In order to gain more insight into the effect of D-serine binding on NMDAR C-terminus properties, we used Fluorescence Lifetime Imaging (FLIM) to measure the Förster Resonance Energy Transfer (FRET) between two GluN1 C-terminal tails, NMDAR obligatory subunits, tagged with fluorescent proteins (*Figure 4A–B*). As previously described (*Dore et al., 2015*), a FRET signal was obtained by co-expressing a 'donor' GluN1-GFP with a GluN1-mcherry 'acceptor' and measured as a decrease in lifetime of the donor fluorescence. We found that a 1:3 ratio provided the best FRET efficiency (GluN1-GFP: 2.52 ± 0.01 ns, $n = 70$ clusters; GluN1-GFP + GluN1-mCherry 1:3 ratio: 2.34 ± 0.01 ns, $n = 78$ clusters; GluN1-GFP + GluN1-mCherry 1:2 ratio: 2.41 ± 0.02 ns, $n = 44$ clusters, $p<0.0001$; *Figure 4B*). Hippocampal neurons were therefore co-transfected with the GluN1 FRET pair at 1:3 ratio, and GluN2B-FLAG to increase synaptic targeting, and imaged 4 days later. As a negative control, we used a mCherry-GluN2B construct whose tag is in the extracellular domain therefore preventing any FRET with the GluN1-GFP (2.50 ± 0.01 ns, $n = 60$ clusters, $p>0.05$ compared to control; *Figure 4B*). We analyzed the FRET signal from NMDAR clusters identified in the fluorescent image, localized both on spines (arrows, *Figure 4B*) and shafts (arrow heads, *Figure 4B*) and found no significant difference in the lifetime between these two regions (spines clusters: 2.32 ± 0.02 ns, $n = 45$; shaft clusters: 2.36 ± 0.02 ns, $n = 43$, $p>0.05$; *Figure 4—figure supplement 1*). In the presence of NMDA (20 µM, 5 min), and as expected from previous reports (*Dore et al., 2015*), the lifetime of GluN1-GFP co-transfected with GluN1-mCherry caused a significant increase in the GluN1-GFP lifetime (NMDA: 2.27 ± 0.01 ns, $n = 124$; Tyrode: 2.23 ± 0.01 ns, $n = 291$, $p<0.05$, *Figure 4C*), indicating a smaller FRET efficiency and validating our ability to measure FRET changes with our system. We next assessed the effect of co-agonist applications on the FRET signal and found that D-serine (30 µM, 5 min) decreased the lifetime of GluN1-GFP co-expressed with GluN1-mCherry (baseline: 2,302 ± 19 ps; D-serine: 2,279 ± 20 ps, $n = 103$, $p<0.05$; *Figure 4D*) while glycine (30 µM, 5 min) had no effect on FRET efficiency (baseline: 2,222 ± 20 ns; glycine: 2,226 ± 19 ps, $n = 110$, $p>0.05$; *Figure 4D*). Although we only detected small changes in the FRET lifetimes after the application of the D-serine, it is tempting to hypothesize that while glycine and D-serine are considered equivalent with regard to their activation of the co-agonist binding site of NMDAR, they have, in fact, a distinct conformational effect on the receptor: D-serine binding specifically elicits a conformational change that brings the two GluN1 C-tails closer (*Figure 4E*) while glycine does not. Interestingly, when the cells were pre-incubated with NMDA (20 µM), the following 5-min incubation with either co-agonist did not produced any further change of the lifetime of the GluN1-GFP (NMDA baseline: 2,309 ± 15 ps; glycine: 2,320 ± 15 ps; NMDA baseline: 2,333 ± 13 ps; D-serine: 2,334 ± 13 ps; $p>0.05$; *Figure 4D*). The NMDA binding conformational change prevails over the D-serine-binding-induced change (−23.2 ± 11 ps lifetime change after D-serine incubation compared to 15.2 ± 9 ps lifetime change after D-serine incubation in the presence of NMDA, $p=0.06$, *Figure 4D*). In the presence of NMDA, the c-terminal tails of the GluN1 subunits probably move apart independently of the co-agonist binding (*Figure 4E*). Our data thus support the conclusion that D-serine binding alone modifies NMDAR conformation in a way that glycine does not; however, the lifetime change detected is only of 23.2 ± 11.8 ps, at this point it is not clear if this change is sufficient to modulate the interactions of NMDAR with their intracellular partners in a co-agonist-dependent manner. Further studies with stronger FRET reporters will surely further establish the relationship between the D-serine binding and the modulation of the C-terminal conformation of NMDAR subunits.

## Co-agonist availability participates to the developmental subunit switch

It was recently suggested that, at CA3-CA1, D-serine gates NMDAR co-agonist binding site in adult (*Yang et al., 2003*; *Papouin et al., 2012*; *Rosenberg et al., 2013*), whereas glycine plays this role at early developmental stages (*Le Bail et al., 2015*). Together with our previous results which suggest

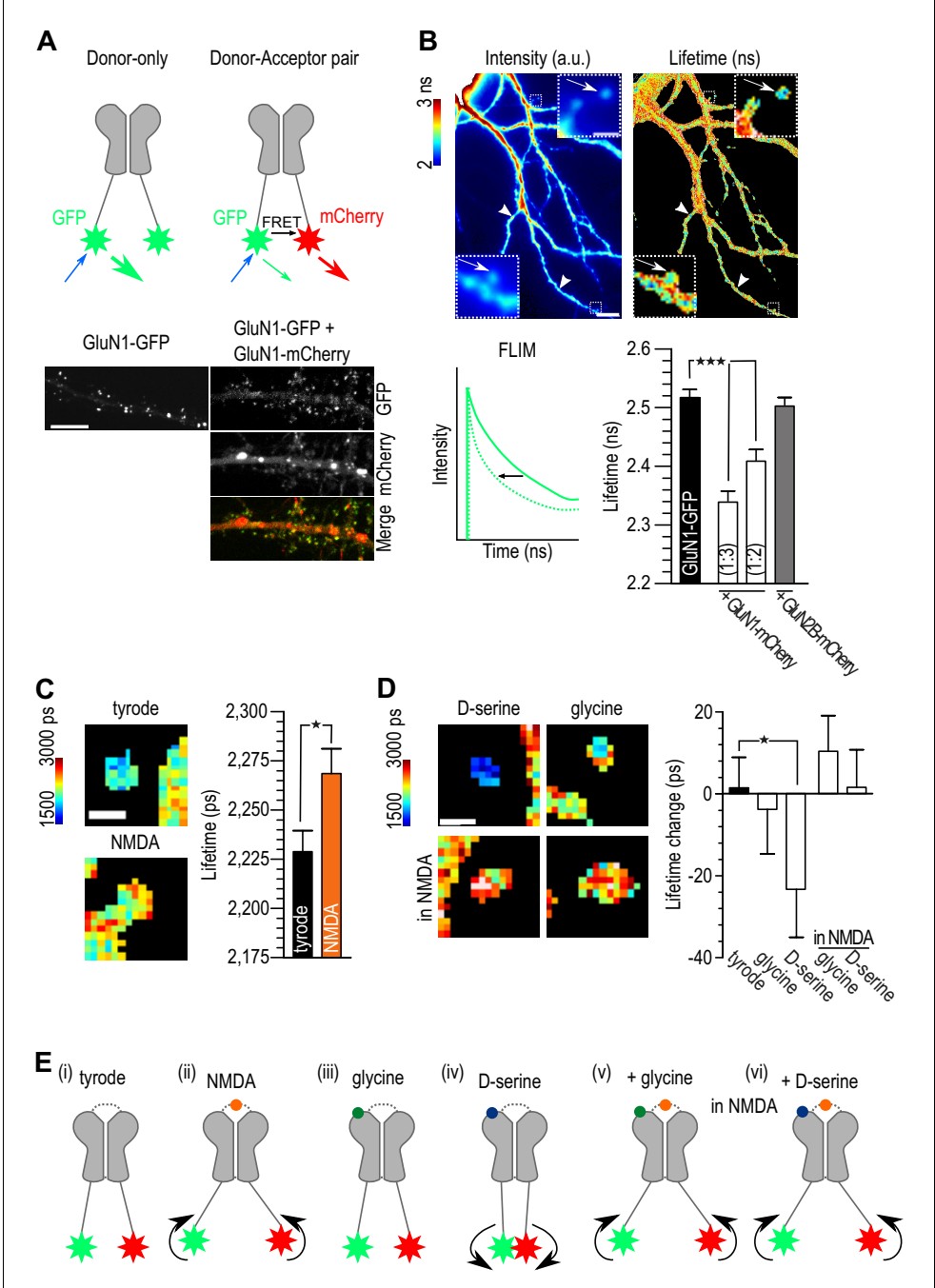

**Figure 4.** D-serine binding alone differently modulated NMDAR c-terminal conformation. (**A**) Schematic representation of experimental design (top). Hippocampal neurons 14 div, transfected at 10 div with: GluN1-GFP (donor) or GluN1-GFP plus GluN1-mCherry (acceptor, bottom). Both conditions were co-transfected with GluN2B-Flag. Scale: 10 μm. (**B**) Example of GluN1-GFP fluorescent (left) and FLIM image (right). NMDAR clusters (spines clusters: arrows; shaft clusters: arrow heads, top) lifetime in nanoseconds (ns) quantification (bottom). Scale: 10 μm, insert 1 μm. Schematic representation of lifetime decay of a donor-only (full line) and of the donor in the presence of the acceptor (dashed line, bottom left). Comparison between GluN1-GFP lifetime alone (donor-only), co-transfected with GluN1-mcherry with a ratio 1:3 or 1:2 (donor-acceptor pair) and co-transfected with GluN2B-mCherry (negative control). GluN1-GFP: n = 70, GluN1-GFP + GluN1-mCherry (1:3): n = 88, GluN1-GFP + GluN1-mCherry (1:2): n = 44, GluN1-GFP + GluN2B-mCherry: n = 60 spines and shaft clusters; p<0.0001, One-way analysis of variance, followed by Dunnett's Multiple Comparison Test, ***p<0.0001 (bottom right). (**c**) Example of FLIM image of GluN1-GFP/GluN1-mCherry clusters after addition of tyrode (control) or NMDA (left). Quantification
*Figure 4 continued on next page*

*Figure 4 continued*

of GluN1-GFP lifetime (right). Tyrode: $n$ = 291, NMDA: $n$ = 124; p=0.0165, $F$ = 1.719, Unpaired t test, one-tail. (**D**) Example of FLIM image of GluN1-GFP/GluN1-mCherry clusters after addition of D-serine or glycine (top) in tyrode only or in the presence of NMDA (bottom). Quantification of GluN1-GFP lifetime change (lifetime after minus the lifetime before co-agonist addition, right). Tyrode: $n$ = 291, glycine: $n$ = 110, D-serine: $n$ = 103, glycine in NMDA: $n$ = 268, D-serine in NMDA: $n$ = 233 spine and shaft clusters; tyrode p=0.4234, $r^2$ = 0.0001, glycine p=0.3650, $r^2$ = 0.0011, D-serine p=0.0255, $r^2$ = 0.0368, glycine in NMDA p=0.0949, $r^2$ = 0.0065, D-serine in NMDA p=0.4348, $r^2$ = 0.0001, Paired t-test, one-tail, before and after, *p<0.05. Data are represented as mean ± s.e.m. (**E**) Schematic representation of the c-terminus tails of the NMDAR in basal conditions (tyrode, (**i**) or in the presence of NMDA (**ii**), the co-agonists alone (**iii, iv**) or in activating conditions (co-agonists together with NMDAR, (**v, vi**).

The following figure supplement is available for figure 4:

**Figure supplement 1.** Lifetime measurement of GluN1-GFP co-transfected 14 div hippocampal neurons, with GluN1-mCherry, and Flag-GluN2B (1:3:2), at 10 div.

that co-agonist availability directly influences the NMDAR-subtype predominating at synapses, we propose that the co-agonist availability could contribute to the subunit developmental switch. GluN2 subunits undergo a developmental switch at hippocampal synapses that makes GluN2B be replaced by GluN2A (**Monyer et al., 1994**; **Sheng et al., 1994**; **Bouvier et al., 2015**). We performed NMDA-fEPSPs recordings (CA1 area) of GluN2A or GluN2B-NMDAR isolated transmission (in presence of Ro25-6981 or zinc, respectively) in slices obtained from postnatal day 5 (P5) to P65. We then bath applied the D-serine or glycine degrading enzymes, RgDAAO and BsGO respectively, to assess the contribution of endogenous D-serine and glycine in allowing NMDAR activation at these different ages (**Figure 5A–C** and **Supplementary file 1**), as in (**Papouin et al., 2012**). By assessing the extend of inhibitory effect of the scavengers on NMDA-fEPSPs, we confirmed that glycine is the main endogenous co-agonist of NMDAR at CA3-CA1 synapses during the first week of postnatal development and gets gradually replaced by D-serine over the second and third weeks of postnatal development (**Figure 5D–F** and **Supplementary file 1**). Therefore, our results predict that these two phenomena are mechanistically linked and that the co-agonist availability influences the NMDAR-subtype predominating at developing synapses.

To test this hypothesis, we assessed whether exogenous applications of D-serine or glycine could change the prevailing NMDAR subtype at CA3-CA1 synapses by measuring the sensitivity of NMDAR-mediated field excitatory post-synaptic potentials (NMDA-fEPSPs) to GluN2A- and GluN2B-NMDAR-specific antagonists (**Paoletti et al., 2013**) zinc (250 nM) and Ro25-6981 (2 µM, **Figure 5G–J**). In slices from young animals (P10-15), when glycine is the major endogenous co-agonist and GluN2B-NMDAR the major NMDAR subtype (**Figure 5A–C**), we observed that applications of exogenous glycine (100–200 µM, 30 min) or D-serine (50–100 µM, 30 min) had no effect on the GluN2A-NMDAR content (control: 37 ± 3.4% inhibition by zinc; glycine: 36.55 ± 3.43%, p<0.01; D-serine: 36.07 ± 3.64%, $n$ = 7, p>0.05; **Figure 5G**) consistent with our finding that GluN2A-NMDAR trafficking remained unaffected by either co-agonist. In slices from adults (>P50), when GluN2A-NMDAR and D-serine predominate, additional D-serine did not modify zinc-induced inhibition but exogenous glycine application elicited a small decrease in GluN2A-NMDAR content (control: 52 ± 2% inhibition by zinc; glycine: 39 ± 2%, p<0.01; D-serine: 51 ± 2% p>0.05, $n$ = 6, **Figure 5H**). Therefore, GluN2A-NMDAR content at synapses was mostly unaffected at either age by the synaptic content of glycine or D-serine, which is consistent with our results (**Figures 1** and **2**). In contrast, the synaptic content of GluN2B-NMDAR in slices from young animals was strikingly reduced when artificially increasing the availability of D-serine, but not glycine (control: 40 ± 3% inhibition by Ro25-6981; glycine: 36 ± 2.5%, p>0.05; D-serine: 22 ± 2%, p<0.01; $n$ = 6–7; **Figure 5I**). Even more strikingly, whereas GluN2B-NMDAR are virtually absent from CA3-CA1 synapses in adult rats under control conditions or after prolonged incubation with D-serine, they contributed to 23 ± 5% of the total synaptic NMDAR population when glycine was exogenously supplied (control: 1.25 ± 2.33% inhibition by Ro25-6981; glycine: 22.65 ± 5.06%, p>0.05; D-serine: 6.36 ± 2.5%, $n$ = 13, p<0.01, **Figure 5J** and **Figure 5—figure supplement 1**). This indicated that providing sufficient amounts of glycine throughout the slice was sufficient to allow GluN2B-NMDAR back in the synaptic space. Notably,

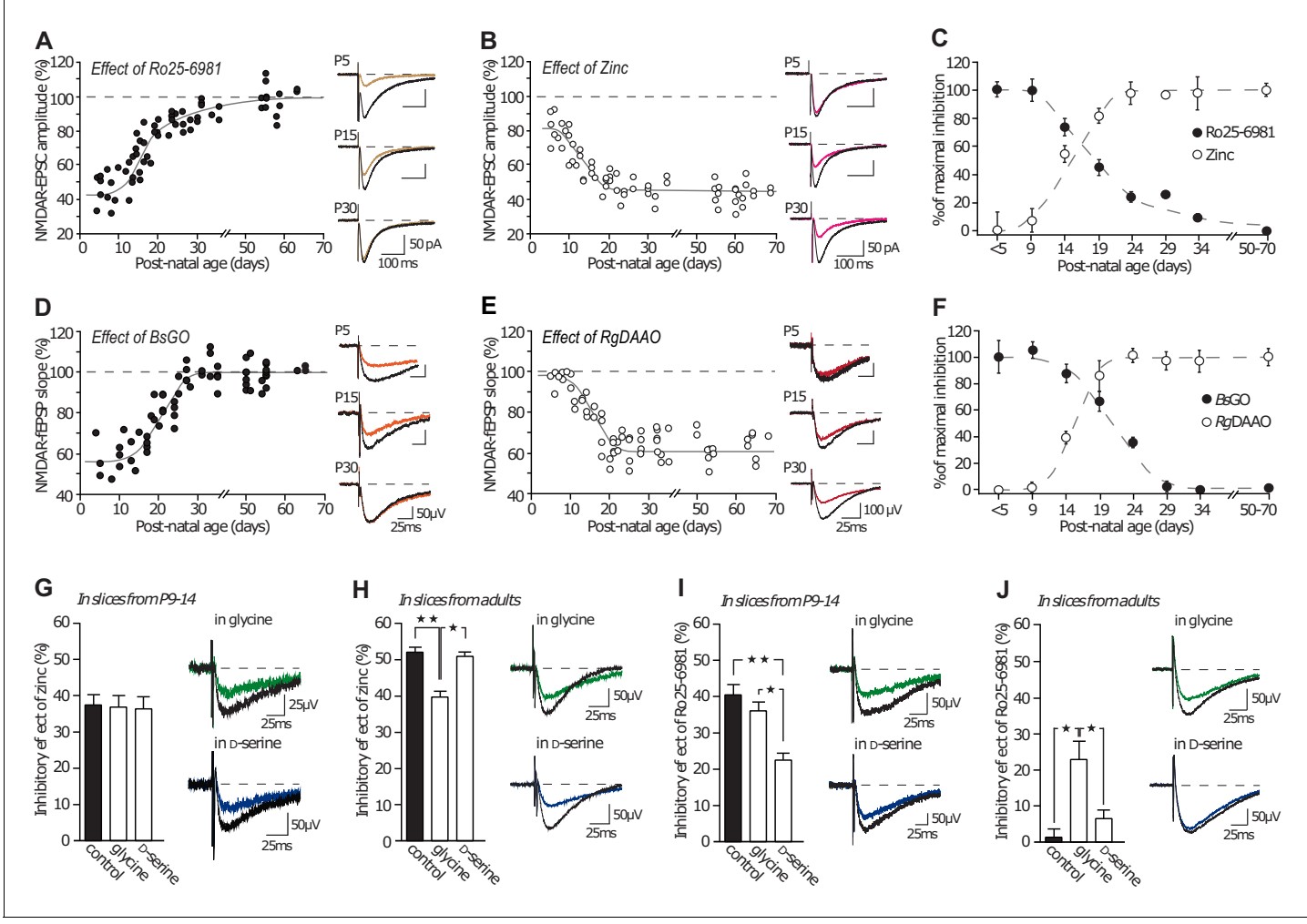

**Figure 5.** GluN2A/GluN2B developmental ratio is differently changed by the co-agonists availability. (A–B) Inhibition mediated by Ro25-6981 (A) and by Zinc (B) on NMDAR-EPSCs in P5 to P70 rats slices. Each dot represents a single experiment. (C) Summary of Ro25-6981 and zinc inhibitory effects throughout development. Data are normalized to the maximal and minimal inhibitory effects and displayed for animals younger than P5 (<P5), from P5-P9 (P9), etc. (D–E) Effects of *Bs*GO (d) and *Rg*DAAO (e) on NMDAR-fEPSPs in P5 to P70 rat slices. (F) Summary of *Bs*GO and *Rg*DAAO inhibitory effects throughout development. Each dot represents a single experiment. *n* and p values can be found in *Supplementary file 1*. (G–H) Inhibitory effects of zinc on NMDAR-fEPSPs in P10-15 rat slices (G) and in adult rat slices (H). P9-P14 slices: *n* = 7, p=0.9319, adults slices: *n* = 6, p=0.0032; Kruskal-Wallis test, followed by Dunn's multiple comparison test *p<0.05, **p<0.01. (I–J) Inhibitory effects of Ro25-6981 on NMDAR-fEPSPs in P10-15 rat slices (I) and in adult rat slices (J). P9-P14 slices: *n* = 6 control and D-serine, *n* = 7 glycine, p=0.0004, adults slices: *n* = 13 control, *n* = 12 glycine, *n* = 11 D-serine; p=0.0003; Kruskal-Wallis test followed by Bonferroni's multiple comparison test *p<0.05, **p<0.01. Data are present as mean ± s.e.m.

The following figure supplements are available for figure 5:

**Figure supplement 1.** NMDAR-EPSC decay time decreases across development.

**Figure supplement 2.** GluN2B subunit does not confer preference for the co-agonist.

D-serine remained without effect on Ro25-6981-mediated inhibition in this experiment indicating that the effect of glycine did not result from the mere recruitment of unsaturated GluN2B-NMDAR. In addition, we previously demonstrated that such application of co-agonist did not alter AMPA receptor synaptic transmission (*Papouin et al., 2012*). In young slices, the change in Ro25-6981-mediated inhibition after D-serine (*Figure 5I*) was not accompanied by a concurrent increase of zinc inhibition (*Figure 5G*). This may be due to the presence of other NMDAR subunits (e.g. GluN2D, GluN3A/B) and/or GluN1/2A/2B triheteromeric receptors, which have different pharmacological

responses than diheteromeric receptors (*Zhu and Paoletti, 2015*). Collectively, these data represent of 'proof of concept' that fluctuations in co-agonist levels, and in particular in the nature of the prevailing co-agonist, markedly alter the NMDAR synaptic content. Intriguingly, these results also strongly advocate that the co-agonist switch occurring during development is instrumental, rather than coincidental, in driving the GluN2B-GluN2A subunit switch. This was further fueled by the finding that glycine levels measured in slices using capillary electrophoresis were ~40% lower in adults than in pups, while D-serine content increased by ~50% over the same period of time (*Figure 5—figure supplement 2B* and *Supplementary file 3*) confirming that the availability of D-serine and glycine changes and reverses throughout development. Finally, to rule out the possibility that the replacement of GluN2B- by GluN2A-NMDAR at synapses determines the preferential use of D-serine over glycine during development, we carried out full dose-response curves for glycine and D-serine on recombinant GluN2A- and GluN2B-NMDAR. For both receptor subtypes, we found that the $EC_{50}$ values for glycine and D-serine as well as the maximal current amplitudes achieved were indistinguishable; indicating that neither subtype of receptor is capable of discriminating between the two co-agonists in respect to binding affinity and extent of activation (*Figure 5—figure supplement 2A* and *Supplementary file 2*). We conclude that the two endogenous co-agonists, glycine and D-serine, i) have the potential to differentially regulate NMDAR function in a subunit-specific manner by eliciting distinct conformational changes that affect the interaction with intracellular partners such as PSD-95, ii) this process is at play in situ, under basal levels of co-agonists, and iii) this mechanism contributes to the NMDAR 'subunit switch' that occurs during postnatal development at the canonical CA3-CA1 synapse.

## Discussion

Understanding the molecular mechanism by which glutamatergic synapses adjust their GluN2A/B-NMDAR signaling has captured a lot of attention over the last decades. Here, we provide evidence that co-agonist availability plays a direct role in the synaptic composition of NMDAR subtypes through the regulation of GluN2B-NMDAR surface dynamics. These data fuel a model in which the developmental up-regulation of D-serine availability at hippocampal synapses reduces the basal surface trafficking of GluN2B-NMDAR, favoring a higher GluN2A/GluN2B-NMDAR ratio at maturing synapses (*Figure 6*). Therefore, we postulate that the NMDAR co-agonists are key regulators of the receptor surface trafficking at hippocampal synapses.

In the central nervous system, the synaptic ratio between GluN2A- and GluN2B-NMDAR is not uniform but varies between developmental stages, brain regions, axonal inputs and sensory

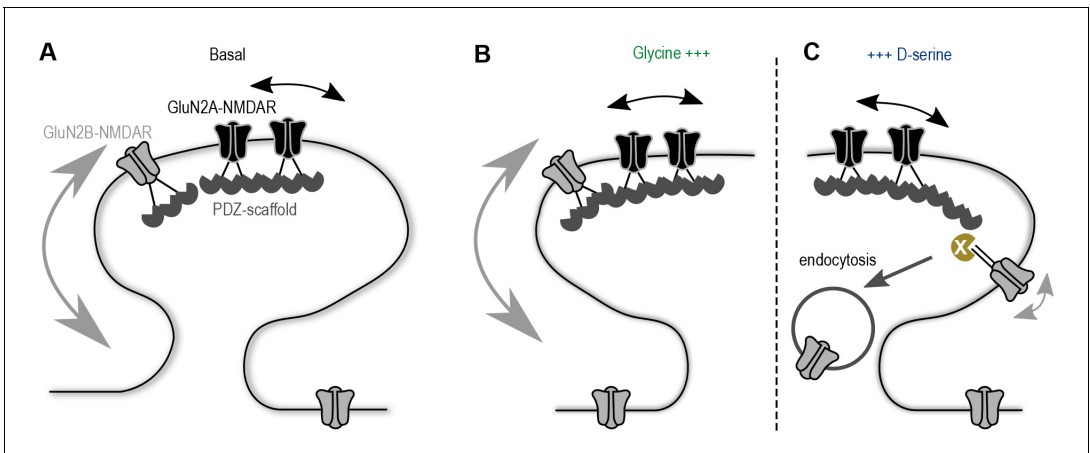

**Figure 6.** Proposed model of the co-agonist differential modulation NMDAR surface dynamics. (A) In basal conditions GluN2A-NMDAR are enriched at synapses, whereas GluN2B-NMDAR are highly mobile. (B) This dynamics is maintained after glycine application. (C) However, D-serine application leads to a specific decrease of GluN2B-NMDAR surface diffusion and synaptic content, through the modulation of the receptors C-terminus interactions, possibly leading to an increase of the receptor internalization.

experience (*Lau and Zukin, 2007*; *Yashiro and Philpot, 2008*). This ratio influences key processes such as synaptic maturation and adult long-term plasticity (*Yashiro and Philpot, 2008*). Therefore, intense efforts have been made to unveil the cellular and molecular mechanisms by which glutamate synapses tune their GluN2A/B subunit ratio. In particular, changes in the postsynaptic scaffold apparatus have been thoroughly investigated during development, providing evidence that PDZ-scaffold proteins (e.g. SAP-102, PSD-95) are differentially involved in the regulation of the GluN2A/B subunit ratio (*Chen et al., 2012*). In addition, extracellular molecules (e.g. reelin, ephrinB) and genetic mechanisms (e.g. transcriptional repressor REST) have also been identified as important regulators of the GluN2-NMDAR switch (*Groc et al., 2007*; *Nolt et al., 2011*; *Rodenas-Ruano et al., 2012*), indicating that multiple molecular cascades are likely involved into this mechanism. Here, we demonstrate that the NMDAR endogenous co-agonists, glycine and D-serine, are also potent regulators of the NMDAR trafficking and synaptic subunit content. This finding is in line with evidence that mice lacking the D-serine synthesizing enzyme serine racemase, as well as mice heterozygotes for the glycine transporter GlyT1 that limits glycine access at excitatory synapses, exhibit altered synaptic GluN2A- and Glu2B-NMDAR content (*Martina et al., 2005*; *Imamura et al., 2008*; *Basu et al., 2009*; *Balu and Coyle, 2011*). In addition, during the induction of long-term synaptic potentiation (LTP) in the hippocampus co-agonists are differentially released (*Henneberger et al., 2010*; *Rosenberg et al., 2013*), likely inducing the fast change in GluN2A/2B-NMDAR ratio (*Bellone and Nicoll, 2007*; *Matta et al., 2011*) associated with a lateral displacement of GluN2B-NMDAR (*Dupuis et al., 2014*). It has been previously shown that the replacement of GluN2B- with GluN2A-NMDAR at synapses is synaptic activity dependent (*Barria and Malinow, 2002*), and require both NMDAR and mGluR5 activation (*Bellone et al., 2011*; *Matta et al., 2011*). Indeed, several mechanisms seem to contribute to this NMDAR subunit switch during development, how do they act in a coordinate manner is still open for investigation.

An intriguing feature of our findings is the striking specificity toward GluN2B-NMDAR. Interestingly, it has been reported that GluN2B-NMDAR surface dynamics are specifically altered when neurons are exposed to the extracellular protein reelin (*Groc et al., 2007*), to sexual hormone estrogen (*Potier et al., 2016*), or to strong synaptic activation (*Dupuis et al., 2014*). These paradigms have in common to be potent regulators of NMDAR-dependent synaptic maturation and plasticity in the hippocampus. Deciphering the molecular mechanism by which D-serine selectively impacts GluN2B-NMDAR conformation, surface trafficking and synaptic retention is a major challenge. Since there seems to be no difference between the binding of glycine and D-serine onto the co-agonist site of NMDAR as far as affinity and efficacy are concerned, one can rule out the possibility that GluN2B-NMDAR use D-serine as a preferred co-agonist. Interestingly, our data favor a model in which the binding of D-serine induces a rapid conformational change of the receptor C-terminus domain and a consequent alteration of the interactions between GluN2B-NMDAR and PDZ scaffold partners. This is further supported by the observations that the activation of NMDAR with the glutamate-site agonist NMDA also induces a conformational change of the receptor that reduces the interaction between the C-terminus and PSD-95 or with protein phosphate 1 (PP1, *Doré et al., 2014*; *Aow et al., 2015*; *Dore et al., 2015*). Noteworthy, the conformational changes specifically induced by D-serine do not alter the ion flux through the receptor, reminiscent of a metabotropic-like effect (*Nabavi et al., 2013*; *Dore et al., 2015*). In-depth investigations of the co-agonist impact on GluN2A and GluN2B subunit conformations will be necessary to shed light on this process.

Our data support a model in which an increase in D-serine levels reduce GluN2B-NMDAR mobility, favor their internalization and down-regulate their synaptic content (*Figure 6*). The decreased interaction between GluN2B-NMDAR and PSD-95 after D-serine incubation may favor the receptor endocytosis by unmasking a GluN2B internalization signal (*Roche et al., 2001*). This molecular cascade is likely not the sole mechanism since the interaction with PDZ scaffolds is required for the receptor internalization, indeed disrupting the GluN2B-NMDAR interactions with PDZ-containing proteins prevents the D-serine-induced synaptic decrease (competing peptide data). This observation implies that there must be some PDZ-containing proteins, interacting directly with GluN2B subunit, which are essential for the receptors displacement. Among the possible candidates, Scrible1 is a synaptic scaffold that contains four PDZ domains. It binds directly to the GluN2B subunit (*Piguel et al., 2014*) and to AP2 adaptor complex, promoting GluN2B-NMDAR internalization (*Scott et al., 2004*; *Prybylowski et al., 2005*). Moreover, D-serine/NMDA stimulation induces GluN2B internalization in the presence of Scrible1, an effect prevented by knocking down the

scaffold protein (*Piguel et al., 2014*). We therefore propose that D-serine-induced C-terminus conformation change favors the interaction of GluN2B subunit with certain PDZ scaffolds (e.g. Scrible1, SAP102) that promote surface diffusion slow-down and internalization. The identification of the specific scaffold proteins and/or MAGUK family responsible for such effect will thus be of great interest.

Overall, beyond the well-established role of the NMDAR co-agonists on the receptor activation, we here provide direct evidence that co-agonists also regulate the receptor trafficking within glutamatergic synapses in a fast and subunit-specific manner. This offers new perspectives for translational research given that NMDAR, glycine and D-serine have been tightly linked to neuropsychiatric disorders, such as schizophrenia (*Balu and Coyle, 2015*). Manipulating either co-agonist levels or the surface dynamics of a NMDAR subtype, at specific developmental stages, may represent new therapeutic opportunities.

## Material and methods

### Enzymes

Recombinant wild-type *R. gracilis* D-amino acid oxidase (*Rg*DAAO, EC 1.4.3.3) and recombinant wild-type *Bacillus subtilis* glycine oxidase (*Bs*GO, EC 1.4.3.19) were overexpressed in *Escherichia coli* cells, purified and used as described earlier (*Pollegioni et al., 1992*; *Sonia et al., 2001*; *Job et al., 2002*; *Papouin et al., 2012*). The final *Rg*DAAO and *Bs*GO preparations had a specific activity of approximatively 75 U/mg protein on D-serine as substrate and 1.1 U/mg protein on glycine as substrate, respectively. When used in combination with electrophysiology experiments, enzymes were simply bath applied (ie added to the solution superfusing the slices), after baseline recordings were obtained, at a final concentration of 0.2 U/ml for 45–60 min until plateau effect was reached.

### Single quantum dot tracking in hippocampal neurons

Hippocampal cultures, containing neurons and glial cells, were prepared from rat at the embryonic stage 18 (E18) and grown on glass coverslips as previously described (*Bard et al., 2010*). Cells 9–17 div were incubated 10 min with 1 µl of polyclonal antibodies against GluN2B or GluN2A subunits (Alomone Labs, Jerusalem, Israel; epitope correspond to residues 323–337 of GluN2B subunit, RRID: AB_2040028, or residues 41–53 of GluN2A subunit, RRID: AB_2040025, 1:200) followed by 10-min incubation with QD 655 Goat F(ab')two anti-rabbit polyclonal antibodies (Invitrogen, Thermo Fisher Scientific Inc., Cambridge, UK, 1:10,000). Cells were then incubated for 30s with 20 nM MitoTracker (Invitrogen, Courtaboeuf, France), which labels the mitochondria (*Li et al., 2004*). In primary hippocampal cultures close to 85% of MitoTracker staining colocalizes with the presynaptic maker bassoon (*Groc et al., 2004*; *Ehlers et al., 2007*). In the experiments, we observed that none of the experimental protocols altered the mitotracker-based synaptic staining (not shown). All incubations were done in Neurobasal Medium supplemented with 1% BSA (SIGMA, Sigma-Aldrich Chemie S.a.r.l., Saint-Quentin Fallavier France) at 37°C. Coverslips were mounted in tyrode solution (30 mM D-glucose, 120 mM NaCl, 5 mM KCl, 2 mM MgCl2, 2 mM CaCl2 and 25 mM HEPES, pH 7.3–7.4) on a heated-chamber for observation. QD were detected by using a mercury lamp and appropriate excitation/emission filters. Images were obtained with an acquisition time of 50 ms (20 Hz) with up to 500 consecutive frames. Signals were detected using an EMCCD camera (Evolve, Photometrics, PHOTOMETRICS,Tucson, AZ). In each recording session, three to four neuronal fields were randomly selected, followed by the application of the co-agonist (glycine or D-serine, 30 µM), or the enzymes (*Bs*GO or *Rg*DAAO, 0.2 U/ml), directly on the imaging chamber, followed by a selection of three to four new neuronal fields. TAT-peptides (*Bard et al., 2010*) [20 µM, TAT-NS: control, scramble sequence; TAT-GluN2B$_{15}$: TAT-sequence (YGRKKRRQRRR) - GluN2B C-terminus sequence (NGH VYEKLSSIESDV)] were incubated 15 min before QD labeling and acquisition. QD recording sessions, which lasted up to 20–25 min, were processed with the Metamorph software (Universal Imaging Corporation, PA, USA). The instantaneous diffusion coefficient 'D' was calculated for each trajectory, from linear fits of the first 4 points of the mean-square-displacement versus time function using MSD (t) =<r2 > (t) = 4 Dt. The two-dimensional trajectories of single molecules in the plane of focus were constructed by correlation analysis between consecutive images using a Vogel algorithm. This technique provides with a high accuracy of single QD detection (~30 nm resolution) which we used to measure the dynamic distribution of GluN2B-NMDAR or GluN2A-NMDAR at synaptic sites. Synaptic

area was defined as the combination of the postsynaptic density labeled with MitoTracker and the perisynaptic zone (surrounding 300 nm area). The synaptic diffusion coefficient was calculated from GluN2-QD trajectories that were only present inside the synaptic area. The synaptic fraction was obtained by calculating the average fraction of QD-coupled GluN2-NMDAR detected inside the synaptic area over the 500 frames of an acquisition. For the comparison of MSD curves between conditions, the MSD values ($\mu m^2$) measured between 0.5 and 0.75s ('$MSD^{0.5-0.75s}$') or 0.35 and 0.55s ('$MSD^{0.35-0.55s}$') time lag were statistically compared (Mann-Whitney test) and expressed as mean ± s.e.m.

## Immunocytochemistry and image analysis

Hippocampal neurons 18–19 days in vitro (div) were pre-incubated 45 min with 30 µM of either glycine or D-serine (control conditions were incubated with tyrode solution only) at 37°C. For the TAT-peptides experiments, hippocampal neurons 16–17 div were incubated with 20 or 2 µM of the TAT-NS or TAT-GluN2B$_{15}$ for 45 min in the presence or absence of 30 µM D-serine. After the treatment, NMDAR were surface stained by live-staining the cells with specific antibodies against the extracellular terminal of GluN2B or GluN2A (homemade antibodies 2 mg/ml, Agro-Bio, La Ferté Saint Aubin, France, 1:200), prepared in conditioned medium, 10 min at 37°C. Cells were fixed in 4% sucrose/4% paraformaldehyde in PBS for 15 min at room temperature, followed by 1-hr incubation with 10% BSA (SIGMA) in PBS to block nonspecific antibody binding. Plasma membrane labeling was detected by staining with anti-rabbit-Alexa 488 (RRID: AB_143165, Invitrogen, Courtaboeuf, France, 1:1000). Homer c1 intracellular staining was used as the synaptic marker. Cells were permeabilized with 0.25% Triton X-100 in PBS, incubated with the primary antibody against Homer (RRID: AB_10549720, Synaptic Systems, Goettingen, Germany, 1:500) followed by secondary incubation with DyLight 594 (Jackson Immuno Research Europe Ltd, Suffolk, UK, 1:1000). To evaluate the efficiency of the TAT disrupting peptides, hippocampal neurons 16 div were incubated for 45 min with 2 µM of TAT-NS or TAT-GluN2B$_{15}$, and live-stained with Anti-GluN2B (as described above). After fixation and permeabilization cells were labeled with Anti-PSD-95 (7E3-1B8, Thermo Scientific, Rockford, 1:1000) followed by secondary incubation with mouse-Alexa 568 (RRID: AB_2534072, Invitrogen, Courtaboeuf, France, 1:1000). For labeling of the total content of GluN2A or GluN2B subunits, cells were fixed with 4% sucrose/4% paraformaldehyde right after stimulation, and permeabilized. Primary antibodies were prepared in 3%BSA-PBS and incubated for 2 hr at room temperature followed by secondary staining as described before. For GluN1 staining, hippocampal neurons were fixed in methanol at −20°C for 10 min as previously described in *Ferreira et al., 2015*), and labeled with anti-GluN1 (54.1, RRID: AB_2533060, Thermo Scientific, Rockford, 1:1500) followed by secondary staining with mouse Alexa 488 (Invitrogen, Courtaboeuf, France, 1:1000). Fluorescence images were acquired using an EMCCD Photometrics Quantem 512 camera and Metamorph imaging software. For each experiment, images in each channel were captured using the same exposure time across all fixed cells; images were acquired as grey scale from individual channels and pseudocolor overlays were prepared using ImageJ. To quantify the immunocytochemistry data, 8–10 cells per condition from each independent experiment were selected. From each neuron, two to three dendrites were chosen for analysis. The digital images were subjected to a user-defined intensity threshold, for clusters selection and background subtraction. Cluster mean intensity was measured for all clusters of the selected region. Synaptic clusters were determined as the postsynaptic clusters overlapping threshold Homer c1. All analyses were done blind to treatment condition.

## Synaptosomes preparation

Synaptosomes were purified by subcellular fractionation of homogenized hippocampus of P30 rats, after incubation with either the saline buffer (control), glycine (30 µM) or D-serine (30 µM) in artificial cerebrospinal fluid (aCSF) during 1 hr (two animals per condition). The aCSF composition was (in mM): 125 NaCl, 2.5 KCl, 1.25 NaH$_2$PO$_4$, 26 NaHCO$_3$ and 10 glucose (pH 7.3, 290–300 mOsm L$^{-1}$). The resulting hippocampi were then collected and frozen with liquid nitrogen. Treated hippocampi were thaw in 3 ml of ice-cold TPS buffer (0.32 M sucrose, 4 mM HEPES pH 7.4) supplemented with a protease inhibitor cocktail (1:1000, Calbiochem, Merck Millipore, Darmstadt, Germany) and homogenized with Teflon glass potter. After centrifugation at 1000 g for 8 min at 4°C, the pellet (P1) was collected and the supernatant (S1) centrifuged once again at 12,500 g for 13 min at 4°C. The

resulting P2 pellet (crude synaptosomes) was resuspended with 1 ml of TPS buffer and layered on top of a two-step sucrose density gradient (0.8 M and 1.2 M prepared in 4 mM Hepes pH 7.4 buffer). After centrifugation at 50,000 g, 4°C for 70 min, synaptosomes were collected from the interface of the two sucrose solutions, protein content quantified.

## Immunoprecipitation

Fifty micrograms of synaptosomes were solubilized in triton buffer (Tris HCl 20 mM pH8, 1.3% triton, EDTA 1 mM) 30 min at 4°C. The antibody against GluN2B (polyclonal Ab, 0.837 mg/mL, 1 µl, F.A. Stephenson, London, UK) or GluN2A (polyclonal Ab, 1 mg/mL, 1 µl, F.A. Stephenson, London, UK) was incubated under constant agitation at 24°C for 15 min with 10 µL of Protein A (Dynabeads Protein A, Invitrogen, Courtaboeuf, France), pre-washed two times with buffer. The antibody-bead mix was added to the solubilized synaptosomes and incubated under constant agitation during 3 hr at 24°C. Beads were then thoroughly washed four times with Triton buffer. The beads were re-suspended in 25 µL of 2x sample buffer. Before loading on a gel, the samples were boiled at 95°C for 5 min.

## Western blot analysis

Synaptosomes samples were prepared with 2x sample buffer and 1 µg of total protein loaded per lane. Before loading, samples were boiled 5 min at 95°C. For immunoprecipitation experiments (IP), 10 µl of the samples without beads were used. Samples were separated by SDS/PAGE (4–20% Mini-PROTEAN TGX Gel Bio-rad, Marnes-la-Coquette, France) for 40 min at 200V. Gels were then transferred onto nitrocellulose membrane during 1 hr at 100V. After blocking 1 hr in 5% milk in Tris-saline - 0.05% tween 20 (TBST), the membranes were hybridized with an anti-GluN2A Ab (1 µg/ml home-made antibody, Agro-Bio), an anti-GluN2B Ab (1 µg/ml home-made antibody, Agro-Bio), an anti-GluN1 Ab (0.25 µg/ml, RRID: AB_396353, BD Biosciences, San Jose, CA) or an anti-synaptophysin-1 Ab (0.05 µg/ml, Synaptic Systems) diluted in TBST 0.5% milk, during 1 hr at room temperature. IP membranes were hybridized with an antibody against GluN2B (RRID: AB_2536210, 1:2000, Rabbit polyclonal Ab, Molecular Probes, Thermo Fisher Scientific Inc., Cambridge, United Kingdom), or a monoclonal antibody against PSD-95 (1:500, Thermo Fisher), or a monoclonal antibody against SAP102 (RRID: AB_2261666, 1:500, NeuroMab, Antibodies Incorporated, Davis, CA). Corresponding secondary antibodies were used at 1:10,000 in TBST 0.5% milk. Detection was performed using the SuperSignal West Femto Maximum Sensitivity Substrate detection kit (Pierce, Thermo Fisher Scientific Inc., Cambridge, UK) revealed with the Chemidoc system (Bio-rad). Quantification of band intensity was performed using Image Lab software (Bio-rad), and NMDAR detection was normalized on the synaptic marker (synaptophysin) detection in each well. Percentage of change in synaptic content of the subunits was calculated as the variation to 100% (no change) of the ratio of the subunit content in glycine divided by the respective content in D-serine, and normalized to its levels in control condition.

## Frequency-domain-based FLIM-FRET

Hippocampal neurons 9–10 div were co-transfected with carboxyl terminally tagged GluN1-GFP and GluN1-mCherry (*Doré et al., 2014*; *Aow et al., 2015*), gift from Paul De Koninck) together with Flag-GluN2B (gift from R. Wenthold) in a proportion 1:3:1, unless stated otherwise. mCherry-GluN2B (pcDNA3, CMV promotor, modified from SEP-GluN2B (*Dupuis et al., 2014*) was used as a FRET-negative control. FRET-FLIM experiments were performed 4 days after calcium phosphate precipitation method (*Vieira et al., 2016*). Briefly, 0.5 µg to 1.5 µg of DNA (per 18 mm coverslips) was mixed with TE (1 mM Tris–HCl pH 7.3, 1 mM EDTA), $CaCl_2$ (2.5 M $CaCl_2$ in 10 mM HEPES, pH 7.2) and 2× HEBS (12 mM dextrose, 50 mM HEPES, 10 mM KCl, 280 mM NaCl and 1.5 mM $Na_2HPO_4 \cdot 2H_2O$, pH 7.2). The precipitates were added to the cells and incubated for 1 hr at 37°C. Both transfection and the following wash were performed in Neurobasal medium containing 2 mM kynurenic acid (SIGMA). Cells were imaged with an HCX PL Apo 100x oil NA 1.4 objective using an appropriate GFP filter set. Cells were excited using a sinusoidally modulated 3 W 478 nm LED (light-emitting diode) at 36 MHz under wild-field illumination. Both the LED and the GenIII image intensifier were modulated at frequency up to 100 MHz. Emission was collected using an intensified CCD LI2CAM camera (Lambert Instruments BV, Groningen, The Netherlands). Lifetimes were calibrated using a solution of

erythrosin B (1 mg/ml) that was set at 0.086 ns. The lifetime of the sample is determined from the fluorescence phase-shift between the sample and the reference from a set of 12 phase settings using the manufacturer's LI-FLIM software. Lifetime was measured in regions defined by the user, in the fluorescent image, blind to the FLIM image, before and after 5-min incubation of the stimuli application (20 µM NMDA, 30 µM D-serine and 30 µM glycine). For control experiments, the appropriate volume of tyrode solution was added. For the experiments performed in the presence of NMDA, 20 µM of NMDA was added 2 min before starting the acquisition.

## Slice preparation

Experiments were carried out on acute hippocampal slices obtained from Wistar rats from 4 to 70 days after birth, at room temperature (20–22°C) in the presence of (in mM) 0.2 to 1.3 $Mg^{2+}$, 2.5 $Ca^{2+}$, 0.05 picrotoxin and 0.01 strychnine. NMDAR-mediated responses were isolated with 10 µM NBQX to block AMPA/Kainate receptors. All experiments were conducted with respect to European and French directives on animal experimentation (authorization no. 33 0004). After decapitation under isoflurane anaesthesia, the brain was quickly removed from the skull and placed in ice-cold aCSF saturated with 95% $O_2$ and 5% $CO_2$. Hippocampal coronal slices (300 µm) were incubated 30 min at 33°C in 2 mM $Mg^{2+}$ and 1 mM $Ca^{2+}$-containing aCSF and then allowed to recover for at least 30 min at room temperature. For electrophysiological recordings, slices were then transferred into a recording chamber, where they were perfused with aCSF (2 ml/min) saturated with 95%$O_2$/5%$CO_2$. A cut between CA3 and CA1 was made to avoid epileptiform activity.

## Patch-clamp recordings

Pyramidal CA1 neurons were identified visually using infrared DIC microscopy (Olympus BX50). Patch clamp recording pipettes (2–4 MΩ) were filled with (in mM): 150 caesium methane-sulfonate; 1.3 $MgCl_2$; 1 EGTA; 10 HEPES; 0.1 $CaCl_2$ (adjusted to pH ~7.2 with CsOH, 290–296 mOsm/kg). Access resistance (Ra) and holding current (Ih) were monitored throughout the experiment. Cells with Ra >25 MΩ or Ih< −150 pA at −65 mV were discarded, as well as cells for which those parameters varied of 20% or more during the recording. NMDAR-mediated excitatory post-synaptic currents (NMDAR-EPSCs) were recorded at +40 mV in the presence of 1.3 mM $Mg^{2+}$ or at −65 mV in the presence of 0.2 mM $Mg^{2+}$.

## Field recordings

Schaffer collaterals were electrically stimulated at 0.05 Hz with a concentric bipolar electrode placed in the *stratum radiatum*. NMDAR-mediated evoked field excitatory post-synaptic potentials (NMDAR-fEPSPs) were recorded in low $Mg^{2+}$ (0.2 mM) using a glass electrode (2–4 MΩ) filled with aCSF and placed in the *stratum radiatum*. Intensity of stimulation (<2 V, 100 µs) was set at ~70% of that triggering population spikes, and the slope of field responses was monitored online. Average EPSCs traces were obtained from at least 10 min of stable recordings and NMDAR-fEPSPs from at least 30 min of stable recording.

## Drugs

The drugs used were picrotoxin 50 µM, strychnine hydrochloride 10 µM, NBQX salt 10 µM (NBQX), glycine 0.1–0.5 mM, d-serine 10 to 100 µM, Ro 25–6981 maleate 2 µM (Ro25-6981) and $ZnCl_2$250 nM used in Tricine 10 mM with the relation $[Zinc]_{free} = [Zinc]_{applied}/200$. Drugs were all purchased from Tocris and bath-applied. It shall be noted that zinc, as well as Ro25-6981, are partial antagonists that inhibit ~70–80% and ~90% of the current flowing through GluN2A- (GluN1/GluN1/GluN2A/GluN2A) and GluN2B-NMDAR (GluN1/GluN1/GluN2B/GluN2B), respectively (*Paoletti et al., 1997*; *Paoletti and Neyton, 2007*).

## Glycine and D-serine dose-response curves

Recombinant NMDAR were expressed in *Xenopus laevis* oocytes after nuclear co-injection of cDNAs (at 10–30 ng/µl) coding for rat GluN1-1a and either rat GluN2A or mouse GluN2B (ratio 1:1). Oocytes were prepared, injected, voltage-clamped, and superfused as described previously (*Paoletti et al., 1997*). Recordings were performed at a holding potential of −60 mV and at room temperature. For dose-response experiments, NMDAR-mediated currents were induced by

simultaneous application of a saturating concentration of glutamate (100 µM) plus increasing concentrations of glycine or D-serine. The external solution contains (in mM): 100 NaCl, 1.5 $BaCl_2$, 2.5 KCl, 5 HEPES, and 0.01 DTPA (pH 7.3). The heavy metal chelator DTPA (diethylenetriamine-penta-acetic acid) was added to prevent tonic inhibition of GluN1/GluN2A receptors by trace amounts of zinc (*Paoletti et al., 1997*). To avoid contamination by endogenous calcium-dependent chloride currents, the oocytes were pre-incubated with 100 µM BAPTA-AM for a minimum of 4 hr prior to testing. Data were collected and analyzed by using pClamp10 (Molecular Devices, Silicon Valley, CA) and fitted using SigmaPlot 10.0 (SSPS) with the following Hill equation: $I_{rel} = 1/(1+(EC_{50}/[A])n_H)$, where $I_{rel}$ is the mean relative current (normalized to the current measured at 100 µM glutamate plus 300 µM glycine or D-serine), [A] the co-agonist concentration and $n_H$ the Hill coefficient. This allowed determining the $EC_{50}$, the half maximal concentration, for each condition.

## Capillary electrophoresis

The amounts of endogenous glutamate, glycine and D-serine in tissue homogenate from acute hippocampal slices of P5 to P70 rats were determined using capillary electrophoresis as described previously (*Fossat et al., 2012*). Two slices from the same animal (~1.5 mg) were incubated in 1 ml oxygenated aCSF for 1 hr, 2 hr after the slicing procedure. Slices were then carefully extracted from the conditioned medium and both were frozen in liquid nitrogen before separate analysis. Tissue samples were first de-proteinized using 5% final cold trichloroacetic acid. Liquid phase of conditioned medium and tissue samples was fluorescently derivatized at room temperature for 2 hr with napthalene-2,3-dicarboxaldehyde (NDA) before being analyzed by capillary electrophoresis with laser-induced fluorescence (CE-LIF) (CE: Beckman Coulter, P/ACE MDQ; LIF: Picometrics, LIF-UV-02, 410 nm, 15 mW) using a hydroxypropyl-$\beta$-cyclodextrin based chiral separation buffer (*Fossat et al., 2012*). All data were collected and analyzed using Karat 32 software v8.0 (Beckman Coulter, Fullerton, CA). Peak identification was made by spiking the fraction with the appropriate amino acid and the quantification of amino acids was made from a standardized curve. The amount of amino acids in slices and conditioned medium was then scaled to the protein content of tissue incubated, determined by the Lowry method using the Pierce BCA protein Assay kit (Thermo Scientific, Courtaboeuf, France) assay with bovine serum albumin as standards.

## Data and statistical analysis

QD recording sessions were processed with Metamorph software (Universal Imaging, Molecular Devices). Immunocytochemistry images were acquired on an inverted confocal spinning-disk microscope (Leica) and analyzed with ImageJ 1.48e. FLIM was performed on an inverted Leica DMI6000B (Leica Microsystem) spinning-disk microscope and using the LIFA frequency domain lifetime attachment (Lambert Instruments, Roden, The Netherlands) and the LI-FLIM software 1.2.12 (Lambert Instruments). The electrophysiological data were recorded with a Multiclamp 700A amplifier (Axon Instruments, Molecular Devices), collected and analyzed using pClamp9 software (Axon Instruments). Average NMDAR-EPSCs traces were fitted with a double exponential and fits with R < 0.950 were discarded. NMDAR-EPSCs decay time was expressed as a weighted Tau, $\tau_w = (A_1\tau_1 + A_2\tau_2)/(A_1 + A_2)$. Data are present as mean ± s.e.m. Statistical analysis was performed using GraphPad Prism 5.03 (GraphPad Software, Inc., La Jolla, CA). Differences between multiple groups, for non-normally distributed data, like single-particle tracking and immunocytochemistry, were analysed by Kruskal-Wallis test, followed by Dunn's Multiple Comparison Test. For normal data, like FLIM-FRET experiments, one-way analysis of variance, followed by Dunnett's multiple comparison test was used. Repeated measures ANOVA followed by Bonferroni's multiple comparison test was used to compare protein levels in Western blots of synaptosomes. Paired or unpaired Student's t test was used for analysis of electrophysiological recordings. Differences between two groups were assessed by Mann-Whitney test, for non-normal data, like TAT-peptide experiments which were compared with the respective control. Lifetime change was analysed by paired Student t-test, comparing before and after drug application per clusters, differences bigger 35% (three times the standard deviation in the control experiments) were eliminated. Significance was assessed at p<0.05, two-tails tests unless otherwise specified in the legends. Symbols used are: *p<0.05; **p<0.001; ***p<0.0001 throughout the manuscript.

## Acknowledgements

This work was supported by grants from INSERM, Agence Nationale pour la Recherche, Conseil Régional d'Aquitaine, NARSAD, Fondation pour la Recherche Médicale, Fédération pour la Recherche sur le Cerveau, and the Human Frontier Science Program. TP, LL and VCL were recipients of studentships from the Ministère de l'Enseignement Supérieur et de la Recherche. JSF was supported by a grant from FCT and POPH/FSE, Portugal. SS and LP were supported by a grant from Fondo di Ateneo per la Ricerca (Università degli Studi dell'Insubria). We thank the laboratory members for the fruitful discussions. The microscopy was done in the Bordeaux Imaging Center, a service unit of the CNRS-INSERM and Bordeaux University, member of the national infrastructure France BioImaging. We are grateful to P De Koninck for GluN1 plasmids (FRET experiments). The help of C Poujol and S Marais is acknowledged.

## Additional information

### Funding

| Funder | Grant reference number | Author |
| --- | --- | --- |
| Agence Nationale de la Recherche | Neuroscience Program | Laurent Groc |

The funders had no role in study design, data collection and interpretation, or the decision to submit the work for publication.

### Author contributions

JSF, Conceptualization, Supervision, Funding acquisition, Validation, Methodology, Writing—original draft, Project administration, Writing—review and editing; TP, Conceptualization, Formal analysis, Validation, Investigation, Visualization, Writing—original draft, Writing—review and editing; LL, Conceptualization, Formal analysis, Validation, Investigation, Methodology, Writing—original draft, Writing—review and editing; AY, Formal analysis, Methodology; VCL, Data curation, Formal analysis, Methodology; DB, JD, J-PM, Data curation, Formal analysis; SS, LP, PP, Resources; SHRO, Investigation, Methodology, Writing—review and editing; LG, Conceptualization, Supervision, Funding acquisition, Validation, Investigation, Writing—original draft, Project administration, Writing—review and editing

### Author ORCIDs

Joana S Ferreira, http://orcid.org/0000-0002-1049-8063
Pierre Paoletti, http://orcid.org/0000-0002-3681-4845
Stéphane Henri Richard Oliet, http://orcid.org/0000-0003-0595-9029
Laurent Groc, http://orcid.org/0000-0003-1814-8145

### Ethics

Animal experimentation: All experiments were carried out in accordance with University of Bordeaux guidelines and regulations. The animal procedures were approved by the ethical committee of the University of Bordeaux (Laurent Groc experimentation authorization number 3306009).

## Additional files

### Supplementary files

• Supplementary file 1. Co-agonist and NMDAR-subunit developmental switch. Summary of the effects of *Bs*GO, *Rg*DAAO, Ro25-6981 and zinc normalized to baseline (mean ± s.e.m) in slices obtained from rats at indicated ages (see *Figure 5—figure supplement 1*). *n* values are indicated as well as p-values (paired Student t-test) assessing the significance of the effect achieved compared to baseline.

• Supplementary file 2. D-serine and glycine dose-response curves. Summary of Hill coefficient ($n_H$), half maximal concentration ($EC_{50}$) and n-values for the glycine/D-serine dose-response curves obtained on GluN2A- and GluN2B-NMDARs recombinants. p-Values (unpaired, two-tailed Student t-test) assess the difference between the $EC_{50}$ obtained with D-serine and glycine.

• Supplementary file 3. Capillary electrophoresis measurements. Summary of the measurements of glutamate, glycine and D-serine content in slices at different ages. Values indicate average content (in nmoles per mg of tissue) ± s.e.m. p-Values indicate the level of significance (student t-test) between values in <P10 slices compared to values in adults.

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
