## [Decision Letter]

Thank you for submitting your article "Co-agonists differentially tune GluN2B-NMDA receptor trafficking at hippocampal synapses" for consideration by *eLife*. Your article has been reviewed by two peer reviewers, and the evaluation has been overseen by a Reviewing Editor and Richard Aldrich as the Senior Editor. The following individuals involved in review of your submission have agreed to reveal their identity: Camilla Bellone (Reviewer #2) and Andrew Plested (Reviewer #3).

The reviewers have discussed the reviews with one another and the Reviewing Editor has drafted this decision to help you prepare a revised submission.

Summary:

In the manuscript, the authors investigate the influence of NMDAR co-agonists, D-serine and glycine, on the relative content of GluN2A and GluN2B. Using complementary approaches, the authors showed that GluN2B-NMDARs surface diffusion is reduced and its motion behavior became confined in presence of D-serine or of a glycine scavenger suggesting that when D-serine prevails, the trafficking of GluN2B containing NMDARs is altered and the synaptic content of GluN2B reduced. This process requires conformational changes modulate the interaction of NMDARs with their intracellular PDZ binding scaffolding partners.

The reviewers felt that this is an impressive study. The authors have deployed a wide range of tools to build a highly plausible investigation of co-agonist action on synaptic content. Whilst some holes remain – see below and no surprise in that given the complexity of the problem – the reviewers felt that the work is a major contribution. The topic is of general interest and the approach is of high quality. They do however feel that the following points must be addressed for being accepted in *eLife*.

First, control experiments are needed to strengthen the conclusions. Those should address the co-localization of Homer 1c and GluN1 to prove that GluN1 has not changed and also examining surface versus the total amount of GluN2A and GluN2B labelling would be important (see below).

Second, rewriting is necessary in particular in the Discussion, addressing the possible implication of triheteromeric receptors or of other NMDAR subunits, as well as the developmental regulations of scaffolding proteins. The authors should discuss the limitations of the study and provide more balance. Particularly here, the small signal size of the FLIM data requires attention.

Essential Revisions:

In Figure 2 by using immunocytochemistry and synaptosome preparation from hippocampi of young rats, the authors show that GluN2B synaptic content is decreased after D-serine incubation. The authors should show the co-localization of Homer 1c and GluN1 to prove that GluN1 has not changed. Moreover, in Figure 2, in order to analyze the surface versus the total amount of GluN2A and GluN2B labeling, the authors should perform a different analysis. First, they need to fix the cells without permeabilization, perform a staining with an Ab recognizing the N-terminal part of the protein, permeabilize the cells and subsequently perform a second staining using an Ab that recognize the C-terminal part of the protein. The immunostaining should also be performed with the scavenger to confirm their results.

The authors show that D-serine reduces the synaptic staining for GluN2B without affecting the total number of GluN2B clusters. In the synaptosome preparation, while the GluN1 and GluN2A levels were not affected by D-Serine application, the GluN2B content was significantly reduced. It is not clear how the authors, based on these results can assume that D-Serine does not change the total amount of NMDARs in the synaptosome. Indeed, since GluN1 homomers cannot be delivered at the membrane, a reduction in GluN2B should be accompanied by an increase in GluN2A content or by an increase in the content of other NMDA receptor subunits. In the Materials and methods, we did not find the explanation for the analysis of the% change in synaptic content. The authors should show the blot of the total homogenate.

To block the interaction between intracellular scaffolding proteins and the C-terminus of GluN2B, the authors used TAT-GluN2B15 peptide mimicking the last 15 aminoacids of GluN2B. There is a discrepancy between the diffusion coefficient of GluN2B in presence of D-Serine between Figure 1 and Figure 3. Since in Figure 3 the diffusion coefficient has been calculated in presence of TAT-NS, it seems that TAT itself can affect the mobility of the receptors. The authors should probably try to use a lower concentration of the TAT. In the immunocytochemistry experiments, the authors should use instead of Homer1c, PSD-95 for the co-localization with the receptors. This experiment will also provide the control for the efficiency of the TAT to block the interaction. In the co-immunoprecipitation experiments, controls are missing. Moreover, the authors should provide the co-ip in the presence of TAT.

The FLIM data are inconclusive. These experiments were done carefully, but unfortunately, the probes that are used are just not at all sensitive – that is, they barely change their lifetime with the conditions. This deficit was seen in the original manuscripts reporting these probes. This fact does not rule out that C-terminal dynamics could be responsible but the statistics bear out that the effects are tiny. In the subsection “D-serine alters the conformation of NMDAR C-terminus”, the authors state that "D-serine.… strongly decreased" with lifetime changing from 2302 ps to 2279 ps! This change is minute. 20ps average changes in lifetime are not meaningful, particularly with a frequency FLIM measurement. These translate into FRET distances in the low Å range, that is, smaller than an individual amino acid. The level of significance reached with a hundred of measurements is quite small and therefore I would not draw conclusions from this section. The question therefore, is what to do with the FRET section. It doesn't add strongly to the conclusions or the mechanism, but doing this experiment was the right thing to do. What is really required here is a FRET pair that is a better reporter (totally beyond the scope of this work). I would favour keeping this data in the paper, but strongly toning down the interpretation, something like:

"Although we could measure small changes in FRET lifetimes, it is not clear whether these changes reflect biologically meaningful interactions among those proteins. More investigations with more robust reporters will be required to strengthen the relationship between D- Serine and intracellular conformational changes." Of course, the other references (e.g. Discussion) to this part should be toned down too.

Please also make more efforts to synthesise all the data. The "subunit switch" as measured here is between 1/3 and 2/3 in magnitude (current). This effect is similar in size to the trafficking effect at its greatest, and the synaptic content as measured by live cell imaging. The effects in biochemistry and with the peptide disrupting PSD95 binding are smaller, which is no surprise. There are parts of the text (FLIM data) where effect size is overstated. Does everything add up? I found it hard to judge. Perhaps a conservative approach is best, but I would like to see the authors approach the question – could D-serine dependent trafficking be the entire story for the subunit switch?

---

## [Author Response]

*Essential Revisions:*

*In Figure 2 by using immunocytochemistry and synaptosome preparation from hippocampi of young rats, the authors show that GluN2B synaptic content is decreased after D-serine incubation. The authors should show the co-localization of Homer 1c and GluN1 to prove that GluN1 has not changed.*

We have now performed a series of immunostaining experiments as suggested by the reviewers. We report that there is no significant change of GluN1 subunit content, strengthening our former conclusion. The new dataset has been included in the revised manuscript (Figure 2—figure supplement 1), and in the section “Synaptic content of GluN2B-NMDAR is reduced by exogenous application of D-serine” of the results. Corresponding protocol can be found in the Materials and methods section.

*Moreover, in Figure 2, in order to analyze the surface versus the total amount of GluN2A and GluN2B labeling, the authors should perform a different analysis. First, they need to fix the cells without permeabilization, perform a staining with an Ab recognizing the N-terminal part of the protein, permeabilize the cells and subsequently perform a second staining using an Ab that recognize the C-terminal part of the protein.*

In order to perform the proposed experiments, one needs to have great antibodies directed against both extra- and intracellular domains of the subunits. Unfortunately, we were not able to find “fully-satisfactory” antibodies that would allow us to do this experiment (e.g. using an antibody directed against the GluN2A subunit C-terminus: low signal-to-noise ratio). However, as an alternative, we used the same antibodies as for the surface staining, but under permeabilizing conditions in order to evaluate the total amount of the receptor. We now show that the total amount (intracellular + membrane) of GluN2A- and GluN2B-NMDAR is unaffected by the presence of co-agonists. These new data have now been included into the revised manuscript (new panels in Figure 2). The corresponding protocols can be found in Materials and methods section.

*The immunostaining should also be performed with the scavenger to confirm their results.*

As suggested, we performed a series of immunostaining with the scavenger enzymes. Since GluN2B-NMDAR diffusion and synaptic content were specifically altered by D-serine, we focused our attention on GluN2B-NMDAR surface staining after incubation with *Bs*GO or *Rg*DAAO. The exposure time was an important parameter to take into account. Indeed, although we were able to detect changes of single GluN2B-NMDAR surface dynamics few minutes after incubation with D-serine, we were not able to detect similar changes using classical immunostaining (e.g. 10 min incubation, data not shown). Changes in synaptic GluN2B-NMDAR with immunostaining were only observed after 30-60 min incubation with D-serine. Thus, we incubated neurons with BsGO or RgDAAO for 45 min. The incubation of neurons with BsGo or RgDAAO did not change the surface content of GluN2B-NMDAR (BsGO: 108 ± 6%, RgDAAO: 107 ± 7% compared to control; control n=50, BsGO n=49, RgDAAO n=50 cells, 5 independent experiments, P=0.7093, Kruskal-Wallis test) nor the synaptic content (GluN2B surface clusters juxtaposed to the Homer1c labeled clusters; BsGO: 115 ± 8%, RgDAAO: 114 ± 8% compared to control, P= 0.5340, Kruskal-Wallis test). At first glance, this outcome may be surprising as we expected a reduction of GluN2B-NMDAR with the enzymes. However, the capacity of the enzymes to scavenge over time the co-agonists is likely not stable, particularly in a living brain cell network. Thus, these experiments do not shed clear lights, as a temporal control of the co-agonist need to be performed with such long incubation with scavenger enzymes. Consequently, we did not include these lines of experiments, as they are not yet informative, in the revised manuscript.

*The authors show that D-serine reduces the synaptic staining for GluN2B without affecting the total number of GluN2B clusters. In the synaptosome preparation, while the GluN1 and GluN2A levels were not affected by D-Serine application, the GluN2B content was significantly reduced. It is not clear how the authors, based on these results can assume that D-Serine does not change the total amount of NMDARs in the synaptosome. Indeed, since GluN1 homomers cannot be delivered at the membrane, a reduction in GluN2B should be accompanied by an increase in GluN2A content or by an increase in the content of other NMDA receptor subunits.*

We agree that it was intriguing that only GluN2B-NMDAR levels were altered in presence of D-serine, with no change in GluN1- and GluN2A-NMDAR. An obvious speculation is that the content of other NMDAR subtypes (not investigated in this study) is altered in these conditions. We have now have mentioned this point in the revised manuscript (subsection “Synaptic content of GluN2B-NMDAR is reduced by exogenous application of D-serine”).

*In the Materials and methods, we did not find the explanation for the analysis of the% change in synaptic content.*

A more detailed description of the analysis of% change in synaptic content has now been added to the Materials and methods in the revised manuscript.

*The authors should show the blot of the total homogenate.*

*To block the interaction between intracellular scaffolding proteins and the C-terminus of GluN2B, the authors used TAT-GluN2B15 peptide mimicking the last 15 aminoacids of GluN2B. There is a discrepancy between the diffusion coefficient of GluN2B in presence of D-Serine between Figure 1 and Figure 3. Since in Figure 3 the diffusion coefficient has been calculated in presence of TAT-NS, it seems that TAT itself can affect the mobility of the receptors. The authors should probably try to use a lower concentration of the TAT.*

The TAT peptides are cell penetrating compounds that are used to deliver big molecules that otherwise would not cross the lipophilic barriers. However, we fully agree that these peptides can have side effects. The 20 µM concentration used for the TAT peptide experiments is based on former evaluations of the binding constants, as well as detrimental effect on the NMDAR surface diffusion (Bard et al., 2010). Here, we report that D-serine effects were “attenuated” in the TAT-NS conditions when compared to the “no peptide” condition, suggesting that the TAT peptide alone affect the receptor trafficking. To tackle this point, we tested the effect of a lower concentration of the TAT peptides (2 µM) on the surface expression of GluN2B subunit, as proposed by the reviewer. Even with such a low concentration the TAT-NS, D-serine effects were “attenuated” when compared to a “no peptide” condition, indicating that the discrepancy between “control” and “TAT-NS” values raised by the reviewer are most likely TAT peptide-induced off-target effects. This set of data has now been included into the revised manuscript (section, “Co-agonist-dependent regulation of GluN2B-NMDAR trafficking depends on interaction between NMDAR and PDZ scaffolds”).

*In the immunocytochemistry experiments, the authors should use instead of Homer1c, PSD-95 for the co-localization with the receptors. This experiment will also provide the control for the efficiency of the TAT to block the interaction.*

Since the TAT peptides disrupt the interaction with several PDZ-containing proteins, not only PSD-95, we used Homer1c, which does not bind directly to the GluN2B subunit, and therefore is potentially less easily modified by the TAT disrupting peptides. Having a stable synaptic marker is obviously important for our quantification method. Nevertheless, we followed the reviewer suggestion and performed additional experiments in which we evaluated the co-localization between surface GluN2B-NMDAR and PSD-95 clusters to assay the efficiency of the TAT-GluN2B15 peptide. Following a 45 min incubation with TAT-NS or TAT-GluN2B15 (2 µM), the number of GluN2B surface clusters that colocalize with PSD-95 clusters was reduced by the TAT-GluN2B15 peptide (TAT-NS incubation, 7.5 ± 0.5 clusters per 10 µm, n= 30 cells, 3 independent experiments; TAT-GluN2B15: 5.8 ± 0.3 GluN2B clusters per 10 µm of dendrite, n= 30 cells; P=0.0091, t-test). These new data have been included in the revised manuscript.

*In the co-immunoprecipitation experiments, controls are missing.*

We have now included the control value in the revised manuscript: “To note, there was no significant difference between control (no agonist) and glycine condition (control: 2.2 ± 0.4, n=4 independent experiments; glycine: 1.9 ± 0.2, n=4; P>0.05, t-test)”.

*Moreover, the authors should provide the co-ip in the presence of TAT.*

As now included into the manuscript (see above), the TAT peptide by itself attenuate the effect of D-Serine on the trafficking of GluN2B-NMDAR. As co-ip experiment are surely less discriminant than live immunocytochemistry and obvious single molecule imaging, we did not perform this experiment as it will not provide better insight into the molecular machinery.

*The FLIM data are inconclusive. These experiments were done carefully, but unfortunately, the probes that are used are just not at all sensitive – that is, they barely change their lifetime with the conditions. This deficit was seen in the original manuscripts reporting these probes. This fact does not rule out that C-terminal dynamics could be responsible but the statistics bear out that the effects are tiny. In the subsection “D-serine alters the conformation of NMDAR C-terminus”, the authors state that "D-serine.… strongly decreased" with lifetime changing from 2302 ps to 2279 ps! This change is minute. 20ps average changes in lifetime are not meaningful, particularly with a frequency FLIM measurement. These translate into FRET distances in the low Å range, that is, smaller than an individual amino acid. The level of significance reached with a hundred of measurements is quite small and therefore I would not draw conclusions from this section. The question therefore, is what to do with the FRET section. It doesn't add strongly to the conclusions or the mechanism, but doing this experiment was the right thing to do. What is really required here is a FRET pair that is a better reporter (totally beyond the scope of this work). I would favour keeping this data in the paper, but strongly toning down the interpretation, something like:*

*"Although we could measure small changes in FRET lifetimes, it is not clear whether these changes reflect biologically meaningful interactions among those proteins. More investigations with more robust reporters will be required to strengthen the relationship between D- Serine and intracellular conformational changes." Of course, the other references (e.g. Discussion) to this part should be toned down too.*

*Please also make more efforts to synthesise all the data.*

We obviously agree with the reviewer that the observed effects are small, although fully significant. Most importantly for FRET experiments, we performed all the adequate control experiments with several constructs that clearly demonstrate our capacity to measure the interaction between GluN1 subunits. In the revised manuscript, we agreed to tune down some of our sentence, without altering our conclusions. We used the proposed sentences of the reviewer, as seen in the subsection “D-serine alters the conformation of NMDAR C-terminus” of the revised manuscript.

*The "subunit switch" as measured here is between 1/3 and 2/3 in magnitude (current). This effect is similar in size to the trafficking effect at its greatest, and the synaptic content as measured by live cell imaging. The effects in biochemistry and with the peptide disrupting PSD95 binding are smaller, which is no surprise. There are parts of the text (FLIM data) where effect size is overstated. Does everything add up? I found it hard to judge. Perhaps a conservative approach is best, but I would like to see the authors approach the question – could D-serine dependent trafficking be the entire story for the subunit switch?*

Our intent was not to claim that D-serine is the entire story for the NMDAR subunit switch. To avoid such a misunderstanding, we have now edited our manuscript, emphasizing the role of other mechanisms in the developmental adaptation of the NMDAR signaling (e.g. role of mGluR5; Discussion, second paragraph).